# A Multi-Grained Self-Interpretable Symbolic-Neural Model For Single/Multi-Labeled Text Classification

**Xiang Hu**[*1], **Xinyu Kong**[1], **Kewei Tu**[*2]
[1] Ant Group [2] ShanghaiTech University

## Abstract

Deep neural networks based on layer-stacking architectures have historically suffered from poor inherent interpretability. Meanwhile, symbolic probabilistic models function with clear interpretability, but how to combine them with neural networks to enhance their performance remains to be explored. In this paper, we try to marry these two systems for text classification via a structured language model. We propose a Symbolic-Neural model that can learn to explicitly predict class labels of text spans from a constituency tree without requiring any access to span-level gold labels. As the structured language model learns to predict constituency trees in a self-supervised manner, only raw texts and sentence-level labels are required as training data, which makes it essentially a general constituent-level self-interpretable classification model. Our experiments demonstrate that our approach could achieve good prediction accuracy in downstream tasks. Meanwhile, the predicted span labels are consistent with human rationales to a certain degree.

## 1 Introduction

Lack of interpretability is an intrinsic problem in deep neural networks based on layer-stacking for text classification. Many methods have been proposed to provide posthoc explanations for neural networks (Lipton, 2018; Lundberg & Lee, 2017; Sundararajan et al., 2017). However, these methods have multiple drawbacks. First, there is only word-level attribution but no high-level attribution such as those over phrases and clauses. Take sentiment analysis as an example, in addition to the ability to recognize the sentiment of sentences, an ideal interpretable model should be able to identify the sentiment and polarity reversal at the levels of words, phrases, and clauses. Secondly, as argued by Rudin (2019), models should be inherently interpretable rather than explained by a posthoc model.

A widely accepted property of natural languages is that *"the meaning of a whole is a function of the meanings of the parts and of the way they are syntactically combined"* (Partee, 1995). Compared with the sequential outputs of layer-stacked model architectures, syntactic tree structures naturally capture features of various levels because each node in a tree represents a constituent span. Such a characteristic motivates us to think about whether the representations of these internal nodes could be leveraged to design an inherently constituent-level interpretable model. One challenge faced by this idea is that traditional syntactic parsers require supervised training and have degraded performance on out-of-domain data. Fortunately, with the development of structured language models (Tu et al., 2013; Maillard et al., 2017; Choi et al., 2018; Kim et al., 2019), we are now able to learn hierarchical syntactic structures in an unsupervised manner from any raw text.

In this paper, we propose a general self-interpretable text classification model that can learn to predict span-level labels unsupervisedly as shown in Figure 1. Specifically, we propose a novel label extraction framework based on a simple inductive bias for inference. During training, we maximize the probability summation of all potential trees whose extracted labels are consistent with a gold label set via dynamic programming

| request_address | navigate | |
| Give me the address | and direct me there. | well-worn situations |
| (a) | | (b) |

Figure 1: Our model can learn to predict span-level labels **without access to span-level gold labels** during training. In examples (a) and (b), only raw texts and sentence-level gold labels {request_address, navigate} and {negative} are given.

---

[*]Corresponding authors. Correspondence to `aaron.hx@antgroup.com` and `tukw@shanghaitech.edu.cn`

Codes available at `https://github.com/ant-research/StructuredLM_RTDT`

with linear complexity. By using a structured language model as the backbone, we are able to leverage the internal representations of constituent spans as symbolic interfaces, based on which we build transition functions for the dynamic programming algorithm.

The main contribution of this work is that we propose a Symbolic-Neural model, a simple but general model architecture for text classification, which has three advantages:

1. Our model has both competitive prediction accuracy and self-interpretability, whose rationales are explicitly reflected on the label probabilities of each constituent.
2. Our model can learn to predict span-level labels without requiring any access to span-level gold labels.
3. It handles both single-label and multi-label text classification tasks in a unified way instead of transferring the latter ones into binary classification problems (Read et al., 2011) in conventional methods.

To the best of our knowledge, we are the first to propose a general constituent-level self-interpretable classification model with good performance on downstream task performance. Our experiment shows that the span-level attribution is consistent with human rationales to a certain extent. We argue such characteristics of our model could be valuable in various application scenarios like data mining, NLU systems, prediction explanation, etc, and we discuss some of them in our experiments.

## 2 PRELIMINARY

### 2.1 ESSENTIAL PROPERTIES OF STRUCTURED LANGUAGE MODELS

Structured language models feature combining the powerful representation of neural networks with syntax structures. Though many attempts have been made about structured language models (Kim et al., 2019; Drozdov et al., 2019; Shen et al., 2021), three prerequisites need to be met before a model is selected as the backbone of our method. Firstly, it should have the ability to learn reasonable syntax structure in an unsupervised manner. Secondly, it computes an intermediate representation for each constituency node. Thirdly, it has a pretraining mechanism to improve representation performance. Since Fast-R2D2 (Hu et al., 2022; 2021) satisfies all the above conditions and also has good inference speed, we choose Fast-R2D2 as our backbone.

### 2.2 FAST-R2D2

Overall, Fast-R2D2 is a type of structured language model that takes raw texts as input and outputs corresponding binary parsing trees along with node representations as shown in Figure 3(a). The representation $e_{i,j}$ representing a text span from the $i_{th}$ to the $j_{th}$ word is computed recursively from its child node representations via a shared composition function, i.e., $e_{i,j} = f(e_{i,k}, e_{k+1,j})$, where $k$ is the split point given by the parser and $f(\cdot)$ is an n-layered Transformer encoder. When $i = j$, $e_{i,j}$ is initialized as the embedding of the corresponding input token. Please note the parser is trained in a self-supervised manner, so no human-annotated parsing trees are required.

## 3 SYMBOLIC-NEURAL MODEL

### 3.1 MODEL

There are two basic components in the Symbolic-Neural model:

1. A Structured LM backbone which is used to parse a sentence to a binary tree with node representations.
2. An MLP which is used to estimate the label distribution from the node representation.

For Structured LMs that follow a bottom-up hierarchical encoding process (such as our default LM Fast-R2D2), context outside a span is invisible to the span, which may make low-level short spans unable to predict correct labels because of a lack of information. So we introduce an optional module to allow information to flow in parse trees from top to down.

The overall idea is to construct a top-down process to fuse information from both inside and outside of spans. For a given span $(i, j)$, we denote the top-down representation as $e'_{i,j}$. We use the Transformer as the top-down encoder function $f'$. The top-down encoding process starts from the root and functions recursively on the child nodes. For the root node, we have $[\cdot, e'_{1,n}] = f'([e_{root}, e_{1,n}])$

where $e_{root}$ is embedding of the special token [ROOT] and $n$ is the sentence length. Once the top-down representation $e'_{i,j}$ is ready, we compute its child representations recursively via $[\cdot, e'_{i,k}, e'_{k+1,j}] = f'([e'_{i,j}, e_{i,k}, e_{k+1,j}])$ as illustrated in Figure 2.

We denote the parameters of the model as $\Psi$, the parameters used in the Structured LM as $\Phi$ and the parameters used in the MLP layer and the top-down encoder as $\Theta$. Thus $\Psi = \{\Phi, \Theta\}$.

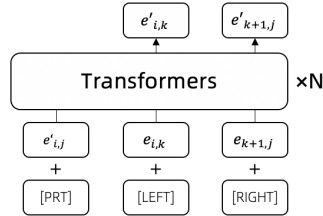

## 3.2 LABEL EXTRACTION FRAMEWORK FOR INFERENCE

During inference, we first use Fast-R2D2 to produce a parsing tree, then predict the label of each node in the parse tree and output a final label set by the **yield** function introduced below.

Figure 2: [PRT], [LEFT], [RIGHT] are role embeddings for the corresponding inputs.

**Inductive bias.** Through observing cases in single/multi-label classification tasks, we propose an inductive bias that a constituent in a text corresponds to at most one label. As constituents could be seen as nodes in a binary parsing tree, we can associate the nodes with labels. Nodes with multiple labels could be achieved by assigning labels to non-overlapping child nodes. Please note such an inductive bias is not applicable for special cases in which a minimal semantic constituent of a text is associated with multiple labels, e.g., the movie "Titanic" could be labeled with both 'disaster' and 'love'. However, we argue that such cases are rare because our inductive bias works well on most single/multi-label tasks as demonstrated in our experiments.

**Label Tree.** A label tree is transferred from a parsing tree by associating each node with a label. A label tree example is illustrated in Figure 3(b). During inference, we predict a probability distribution of labels for each node and pick the label with the highest probability. To estimate the label distribution, we have $P_{\Psi}(\cdot|n_{i,j}) = \mathrm{softmax}(\mathrm{MLP}(e_{i,j}))$. Please note if the top-down encoder is enabled, we replace $e_{i,j}$ with $e'_{i,j}$.

---

**Algorithm 1** Definition of Yield function

1: **function** YIELD($\hat{t}$)
2:     $\mathcal{S} = \{\}$
3:     $q \leftarrow [\hat{t}.root]$
4:     ▷ The list of nodes to visit
5:     **while** $\mathrm{len}(q) > 0$ **do**
6:         $n_{ij} \leftarrow q.\mathrm{pop}(0)$
7:         **if** $n_{ij}.$label $== \phi_{NT}$ **then**
8:             **if** not $n_{ij}.$is_leaf **then**
9:                 $q.\mathrm{append}(n_{ij}.\mathrm{left})$
10:                $q.\mathrm{append}(n_{ij}.\mathrm{right})$
11:         **else**
12:             $\mathcal{S} = \mathcal{S} \cup \{n_{ij}.\mathrm{label}\}$
13:     $\mathcal{S} = \mathcal{S} \setminus \{\phi_T\}$
14:     **return** $\mathcal{S}$

---

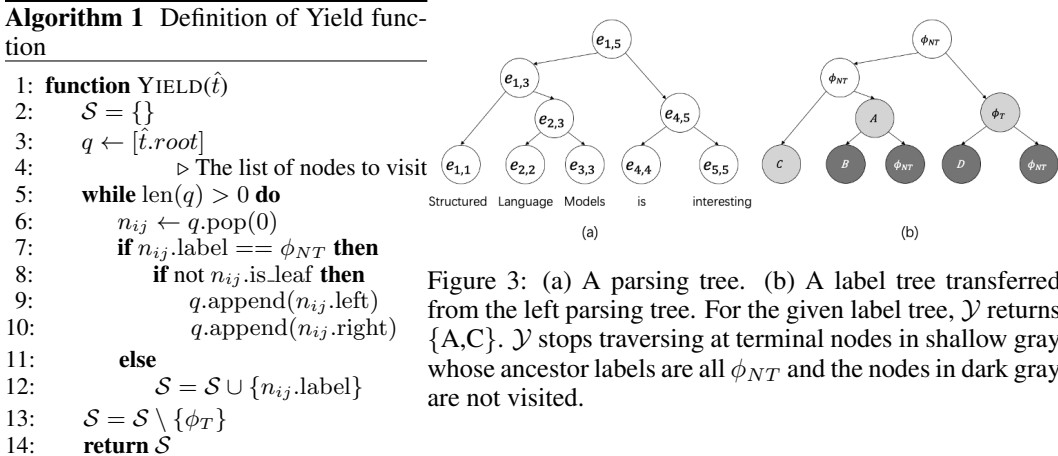

Figure 3: (a) A parsing tree. (b) A label tree transferred from the left parsing tree. For the given label tree, $\mathcal{Y}$ returns {A,C}. $\mathcal{Y}$ stops traversing at terminal nodes in shallow gray whose ancestor labels are all $\phi_{NT}$ and the nodes in dark gray are not visited.

**Yield function.** We design a **yield** function that traverses a label tree in a top-down manner and extracts labels. For brevity, we use $\mathcal{Y}$ short for the **yield** function. We divide the labels into two categories: terminal labels and non-terminal labels, which indicate whether $\mathcal{Y}$ should stop or continue respectively when it traverses to a node. Considering some nodes may not be associated with any task-defined labels, we introduce empty labels denoted as $\phi_T$ and $\phi_{NT}$ for terminal and non-terminal ones respectively. For simplicity, we do not discuss nesting cases[1] in this paper, so there is only one unique non-terminal label which is $\phi_{NT}$ and all task-defined labels are terminal labels. However, our method can be naturally extended to handle nesting cases by allowing non-terminal labels to be associated with task labels. As defined by the pseudo-code in Algorithm 1, $\mathcal{Y}$ traverses a label tree from top to down starting with the root; when it sees $\phi_{NT}$, it continues to traverse all

---

[1]For example, in aspect-based sentiment analysis, a span corresponding to the sentiment may be nested in a span corresponding to the aspect.

its children; otherwise, when it sees a terminal label, it stops and gathers the task-defined terminal label of the node. Figure 3 illustrates how $\mathcal{Y}$ traverses the label tree and gathers task-defined labels.

## 3.3 Training objective.

During the training stage, though the Structured LM can predict tree structures, the difficulty here is how to associate each node with a single label without span-level gold labels. We define our training objective as follows:

**Training objective**    Given a sentence $\mathbf{S}$ whose length is $|\mathbf{S}|$ and its gold label set $\mathcal{T} = \{l_1, ..., l_m\}$, $t$ is its best parsing tree given by the unsupervised parser of Fast-R2D2 and $\hat{t}$ is a label tree transferred from $t$. $\hat{t}^{[\mathcal{C}]}$ denotes $\hat{t}$ satisfying condition $\mathcal{C}$. The training objective is to maximize the probability of a given tree transferring to a label tree yielding labels that are consistent with the ground-truth labels, which could be formalized as minimizing $-\log P_{\Psi}(\hat{t}^{[\mathcal{Y}(\hat{t})=\mathcal{T}]}|t)$.

Before we get into the specifics, several key aspects are defined as follows:

(1) **Denotations**: $t_{i,j}$ denotes the subtree spanning from $i$ to $j$ (both indices are inclusive), whose root, left and right subtree are $n_{i,j}$, $t_{i,k}$ and $t_{k+1,j}$ respectively in which $k$ is the split point.

(2) **Symbolic Interface**: $P_{\Psi}(l|n_{i,j})$ is the probability of a single node $n_{i,j}$ being associated with the specified label $l$. Thus, the probability of $t$ transferring to a specific label tree $\hat{t}$ is the product of all the probabilities of nodes being associated with the corresponding labels in $\hat{t}$.

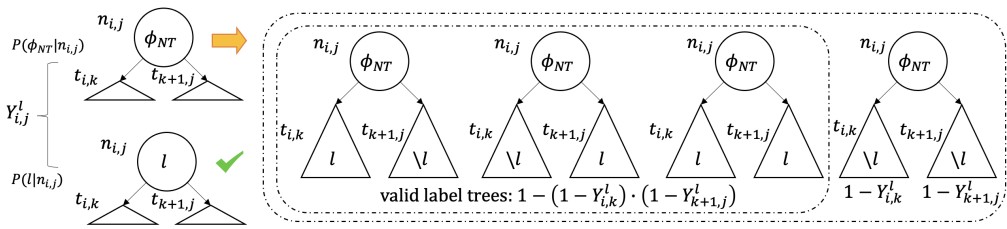

Figure 4: To ensure that the yield result of $\hat{t}_{i,j}$ contains label $l$, node $n_{i,j}$ needs to be associated with either $\phi_{NT}$ or $l$, whose probabilities are $P_{\Psi}(\phi_{NT}|n_{i,j})$ and $P_{\Psi}(l|n_{i,j})$ respectively. If associated with $l$, it satisfies the condition. If associated with $\phi_{NT}$, at least one of its children's yield results should contain $l$. Here we use $\backslash l$ to denote that the yield result does not contain label $l$. In conclusion, $Y_{i,j}^l$ could be estimated recursively by Equation 1.

Obviously, it is intractable to exhaust all potential $\hat{t}$ to estimate $P_{\Psi}(\hat{t}^{[\mathcal{Y}(\hat{t})=\mathcal{T}]}|t)$. Our core idea is to leverage symbolic interfaces to estimate $P_{\Psi}(\hat{t}^{[\mathcal{C}]}|t)$ via dynamic programming. We start with an elementary case: estimate the probability that the yield result of $t_{i,j}$ contains a given label $l$, i.e., $P_{\Psi}(\hat{t}_{i,j}^{[l \in \mathcal{Y}(\hat{t}_{i,j})]}|t_{i,j})$. For brevity, we denote it as $Y_{i,j}^l$. As the recursive formulation illustrated in Figure 4, we have:

$$Y_{i,j}^l = \begin{cases} P_{\Psi}(l|n_{i,j}) + P_{\Psi}(\phi_{NT}|n_{i,j}) \cdot (1 - (1 - Y_{i,k}^l) \cdot (1 - Y_{k+1,j}^l)) & \text{if } i < j \\ P_{\Psi}(l|n_{i,j}) & \text{if } i = j \end{cases} \quad (1)$$

However, for a given label set $\mathcal{M}$, if we try to estimate $P_{\Psi}(\hat{t}_{i,j}^{[\mathcal{Y}(\hat{t}_{i,j})=\mathcal{M}]}|t_{i,j})$ in the same way, we will inevitably exhaust all potential combinations as illustrated in Figure 5(a) which will lead to exponential complexity.[2]

To tackle the problem of exponential complexity, we try to divide the problem of estimating $P_{\Psi}(\hat{t}_{i,j}^{[\mathcal{Y}(\hat{t})=\mathcal{M}]}|t_{i,j})$ to estimating $Y_{i,j}^l$ for each label $l$ in $\mathcal{M}$. Let $\mathcal{F}$ denote the set union of all the task labels and $\{\phi_T, \phi_{NT}\}$, and let $\mathcal{O}$ denote $\mathcal{F} \setminus \mathcal{T}$. By assuming that the states of labels are independent of each other, where the state of a label indicates whether the label is contained in the

---

[2]Details about the dynamic programming algorithm with exponential complexity to estimate $P_{\Psi}(\hat{t}^{[\mathcal{Y}(\hat{t})=\mathcal{T}]}|t)$ is included in Appendix A.2.

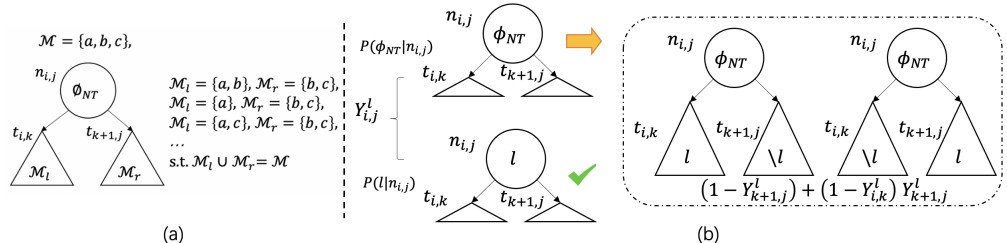

Figure 5: (a) Potential valid yield results of left and right children for $\mathcal{M} = \{a, b, c\}$. (b) Valid label trees when we include a mutual-exclusiveness constraint.

yield result[3], we have:

$$P_\Psi(\hat{t}^{[\mathcal{Y}(\hat{t})=\mathcal{T}]}|t) = P_\Psi(\hat{t}^{[\mathcal{T}\subseteq\mathcal{Y}(\hat{t})]}, \hat{t}^{[\mathcal{O}\cap\mathcal{Y}(\hat{t})=\phi]}|t) \approx P_\Psi(\hat{t}^{[\mathcal{T}\subseteq\mathcal{Y}(\hat{t})]}|t) \cdot P_\Psi(\hat{t}^{[\mathcal{O}\cap\mathcal{Y}(\hat{t})=\phi]}|t)$$

$$P_\Psi(\hat{t}^{[\mathcal{T}\subseteq\mathcal{Y}(\hat{t})]}|t) \approx \prod_{l\in\mathcal{T}} P_\Psi(\hat{t}^{[l\in\mathcal{Y}(\hat{t})]}|t) = \prod_{l\in\mathcal{T}} Y_{i,j}^l \,,\, P_\Psi(\hat{t}^{[\mathcal{O}\cap\mathcal{Y}(\hat{t})=\phi]}|t) = 1 - P_\Psi(\hat{t}^{[\mathcal{O}\cap\mathcal{Y}(\hat{t})\neq\phi]}|t) \quad (2)$$

We do not approximate $P_\Psi(\hat{t}^{[\mathcal{O}\cap\mathcal{Y}(\hat{t})\neq\phi]}|t)$ as it could be computed directly. The above function premises that multiple non-overlapping spans could associate with the same label. In some cases, if there is a mutual-exclusiveness constraint that two non-overlapping spans are not allowed to associate with the same task label as shown in Figure 5(b), the function becomes:

$$Y_{i,j}^l = \begin{cases} P_\Psi(l|n_{i,j}) + P_\Psi(\phi_{NT}|n_{i,j}) \cdot (Y_{i,k}^l \cdot (1 - Y_{k+1,j}^l) + Y_{k+1,j}^l \cdot (1 - Y_{i,k}^l)) & \text{if } i < j \\ P_\Psi(l|n_{i,j}) & \text{if } i = j \end{cases} \quad (3)$$

Regarding $P_\Psi(\hat{t}^{[\mathcal{O}\cap\mathcal{Y}(\hat{t})\neq\phi]}|t)$, $\mathcal{Y}(\hat{t})$ containing any label $l \in \mathcal{O}$ would satisfy the condition. We denote it as $Y_{i,j}^{\mathcal{O}}$ in short. Similar to Equation 1 we have:

$$Y_{i,j}^{\mathcal{O}} = \begin{cases} \sum_{l\in\mathcal{O}} P_\Psi(l|n_{i,j}) + P_\Psi(\phi_{NT}|n_{i,j}) \cdot (1 - (1 - Y_{i,k}^{\mathcal{O}}) \cdot (1 - Y_{k+1,j}^{\mathcal{O}})) & \text{if } i < j \\ \sum_{l\in\mathcal{O}} P_\Psi(l|n_{i,j}) & \text{if } i = j \end{cases} \quad (4)$$

Thus $P_\Psi(\hat{t}^{[\mathcal{Y}(\hat{t})=\mathcal{T}]}|t) = \prod_{l\in\mathcal{T}} Y_{1,|\mathbf{S}|}^l \cdot (1 - Y_{1,|\mathbf{S}|}^{\mathcal{O}})$ and the objective function given a parsing tree is:

$$\mathcal{L}_{cls}^t(\Psi) = -\log P_\Psi(\hat{t}^{[\mathcal{Y}(\hat{t})=\mathcal{T}]}|t) = -\sum_{l\in\mathcal{T}} \log Y_{1,|\mathbf{S}|}^l - \log(1 - Y_{1,|\mathbf{S}|}^{\mathcal{O}}) \quad (5)$$

Because it has been verified in prior work Hu et al. (2022) that models could achieve better downstream performance and domain-adaptivity by training along with the self-supervised objective $\mathcal{L}_{self}(\Phi)$, we design the final loss as follows:

$$\mathcal{L} = \mathcal{L}_{cls}^t(\Psi) + \mathcal{L}_{self}(\Phi) \quad (6)$$

# 4 EXPERIMENTS

## 4.1 DOWNSTREAM TASKS

In this section, we compare our interpretable symbolic-Neural model with models based on dense sentence representation to verify our model works as well as conventional models. All systems are trained on raw texts and sentence-level labels only.

**Data set.** We report the results on the development set of the following datasets: SST-2, CoLA (Wang et al., 2019), ATIS (Hakkani-Tur et al., 2016), SNIPS (Coucke et al., 2018), StanfordLU (Eric et al., 2017). Please note that SST-2, CoLA, and SNIPS are single-label tasks and ATIS, StanfordLU are multi-label tasks. There are three sub-fields in StanfordLU including navigator, scheduler, and weather.

**Baselines.** To fairly compare our method with other systems, all backbones such as Fast-R2D2 and BERT (Devlin et al., 2019) are pretrained on the same corpus with the same vocabulary and epochs. We record the best results of running with 4 different random seeds and report the mean of

---

[3]Details about the conditional independence assumption could be found in Appendix A.8

them. Because of GPU resource limit and energy saving, we pretrain all models on Wiki-103 (Merity et al., 2017), which contains 110 million tokens[4]. To compare our model with systems only using whole sentence representations, we include BERT and Fast-R2D2 using root representation in our baselines. To study the reliability of the unsupervised parser, we include systems with a supervised parser Zhang et al. (2020) that uses BERT or a tree encoder as the backbone. For the former, we take the average pooling on representations of words in span (i,j) as the representation of the span. For the latter, we use the pretrained R2D2 tree encoder as the backbone. To compare with methods dealing with multi-instance learning (MIL) but without structure constraints, we extend the multi-instance learning framework proposed by Angelidis & Lapata (2018) to the multi-instance multi-label learning (MIMLL) scenario. Please find the details about the MIL and MIMLL in Appendix A.7. We also conduct ablation studies on systems with or without the top-down encoder and the mutual-exclusiveness constraint. For the systems using root or [CLS] representations on multi-label tasks, outputs are followed by a sigmoid layer and filtered by a threshold that is tuned on the training set.

**Hyperparameters.** Our BERT follows the setting in Devlin et al. (2019), using 12-layer Transformers with 768-dimensional embeddings, 3,072-dimensional hidden layer representations, and 12 attention heads. The setting of Fast-R2D2 follows Hu et al. (2022). Specifically, the tree encoder uses 4-layer Transformers with other hyper-parameters same as BERT and the top-down encoder uses 2-layer ones. The top-down parser uses a 4-layer bidirectional LSTM with 128-dimensional embeddings and 256-dimensional hidden layers. We train all the systems across the seven datasets for 20 epochs with a learning rate of $5 \times 10^{-5}$ for the encoder, $1 \times 10^{-2}$ for the unsupervised parser, and batch size 64 on 8 A100 GPUs.

| Backbone Multi-Label% | Arch. | #Param. | SST-2 0 | CoLA 0 | SNIPS 0 | ATIS 1.69 | Nav. 26.54 | Sche. 24.86 | Wea. 5.39 |
|---|---|---|---|---|---|---|---|---|---|
| BERT(Wiki-103) | Sent. | 116M | 89.54 | 34.99 | 98.86 | 98.56 | 89.25 | **94.14** | 96.98 |
| parser+BERT | S.N. | 172M | 89.11 | 9.46 | 99.00 | 93.50 | 80.06 | 81.77 | 95.22 |
| parser+TreeEnc. | Sent. | 128M | 88.53 | 11.77 | 99.04 | 97.52 | 89.50 | 93.74 | 96.98 |
| parser+TreeEnc. | S.N. | 128M | 88.19 | 21.49 | 98.93 | 95.77 | 88,94 | 86.95 | 96.50 |
| Fast-R2D2 | MIL | 62M | 89.22 | 34.05 | 98.66 | - | - | - | - |
| Fast-R2D2 | MIMLL | 62M | - | - | - | 93.29 | 84.21 | 89.66 | 94.39 |
| Fast-R2D2 | Sent. | 62M | **89.96** | 36.31 | 99.00 | 98.11 | 89.25 | 93.68 | 96.70 |
| Fast-R2D2 | S.N.$_{fp}$ | 62M | - | - | - | 98.61 | 89.38 | 91.18 | 96.96 |
| Fast-R2D2$_{topdown}$ | S.N.$_{fp}$ | 67M | - | - | - | 98.45 | 88.91 | 93.46 | 97.13 |
| Fast-R2D2 | S.N. | 62M | 89.91 | 35.02 | 98.86 | **98.78** | 88.18 | 91.94 | **97.43** |
| Fast-R2D2$_{exclusive}$ | S.N. | 62M | 89.45 | **36.68** | 99.00 | 98.27 | 87.86 | 89.87 | 96.47 |
| Fast-R2D2$_{topdown}$ | S.N. | 67M | 89.45 | 36.49 | **99.14** | 98.50 | **90.82** | 93.47 | 97.21 |
| Fast-R2D2$_{top./excl.}$ | S.N. | 67M | 89.91 | 35.31 | 99.14 | 98.16 | 90.75 | 93.80 | 96.94 |

Table 1: We report mean accuracy for SST-2, Matthews correlation for CoLA, and F1 scores for the rest. We use "S.N." to denote the systems based on the Symbolic-Neural architecture, and "Sent." to denote those using only whole sentence representations. We use subscript $fp$ for the models based on full permutation, $topdown$, and $exclusive$ for those with the top-down encoder and the mutual-exclusiveness constraint. Please find the details of $S.N._{fp}$ in Appendix A.2.

**Results and discussion.** We make several observations from Table 1. Firstly, We find that our models overall achieve competitive prediction accuracy compared with strong baselines including BERT, especially on multi-label tasks. The result validates the rationality of our label-constituent association inductive bias. The significant gap compared to MIMLL fully demonstrates the superiority of building hierarchical relationships between spans in the model. Secondly, when using sentence representation, the models with the unsupervised parser achieve similar results to those with the supervised parser on most tasks but significantly outperform the latter on CoLA. A possible reason for the poor performance of the latter systems on CoLA is that there are many sentences with grammar errors in the dataset which are not covered by the training set of the supervised parser. While the unsupervised parser can adapt to those sentences as $\mathcal{L}_{bilm}$ and $\mathcal{L}_{KL}$ are included in the final loss. The result reflects the flexibility and adaptability of using unsupervised parsers. Thirdly, 'parser+TreeEnc.' in Symbolic-Neural architectures does not perform as well as 'parser+TreeEnc.'

---

[4]Our model could also be tuned based on the public version of pretrained Fast-R2D2 which is available at `https://github.com/alipay/StructuredLM_RTDT/releases/tag/fast-R2D2`. Details for pretraining BERT and Fast-R2D2 from scratch used in this paper could be found in Appendix A.3

using sentence representation, while the systems using the unsupervised parser show opposite results. Considering that the Symbolic-Neural model relies heavily on the representation of inner constituents, we suppose such results ascribe to the tree encoder having adapted to the trees given by the unsupervised parser during the pretraining stage of Fast-R2D2, which leads to the self-consistent intermediate representations. This result also verifies the structured language model that learns latent tree structures unsupervisedly is mature enough to be the backbone of our method.

## 4.2 ANALYSIS OF INTERPRETABILITY.

Bastings et al. (2022) propose a method that "poisons" a classification dataset with synthetic shortcuts, trains classifiers on the poisoned data, and then tests if a given interpretability method can pick up on the shortcut.

**Setup.** Following the work, we define two shortcuts with four continuous tokens to access the faithfulness of predicted span labels: #0#1#2#3 and #4#5#6#7 indicate label 1 and 0 respectively. We select SST2 and CoLA as the training sets, with additional 20% synthetic data. We create a synthetic example by (1) randomly sampling an instance from the source data, (2) inserting the continuous tokens at random positions, and (3) setting the label as the shortcut prescribes.

**Verification steps.** The model trained on the synthesis data could achieve 100% accuracy on the synthetic test data and the model trained on the original dataset achieves around 50% on the synthetic test set.

**Sorting tokens.** Since our model does not produce a heatmap for input tokens, it lacks an intuitive way to get top K tokens as required in the shortcut method. So we propose a simple heuristic tree-based ranking algorithm. Specifically, for a given label, we start from the root denoted as $n$ and compare $P(l|n_{left})$ and $P(l|n_{right})$ where $n_{left}$ and $n_{right}$ are its left and right children. If $P(l|n_{left}) > P(l|n_{right})$, all descendants of the left node are ordered before the descendants of the right child, and vice versa. By recursively ranking according to the above rule, we could have all tokens ranked. We additionally report the precision of shortcut span labels in the predicted label trees. A shortcut span label is correct only if the continuous shortcut tokens are covered by the same span and the predicted label is consistent with the shortcut label.

**Results.** From the table 2, we have an interesting finding that the precision declines with the increase of training epochs. We think the reason for this phenomenon is that shortcut spans are the easiest to learn at early epochs, so almost all top tokens are short-cut tokens. With continuous training, the model gradually learns the semantics of texts from the original data. Although the label for a sentence in the synthetic data is random, there is still around 50% probability that it is semantically consistent with the text and hence the label probability of a certain span may exceed the

| Models | epochs | top4 prec. | | span label prec. | |
|---|---|---|---|---|---|
| | | SST2 | CoLA | SST2 | CoLA |
| Symbolic-Neural | 1 | 93.31% | 99.11% | 100% | 100% |
| Symbolic-Neural | 3 | 95.10% | 99.30% | 100% | 100% |
| Symbolic-Neural | 5 | 93.46% | 98.95% | 100% | 100% |
| Symbolic-Neural | 7 | 93.23% | 98.25% | 100% | 100% |
| Symbolic-Neural | 9 | 87.44% | 98.83% | 100% | 100% |
| Symbolic-Neural$_{topdown}$ | 1 | 97.31% | 99.23% | 100% | 100% |
| Symbolic-Neural$_{topdown}$ | 3 | 98.76% | 99.95% | 100% | 100% |
| Symbolic-Neural$_{topdown}$ | 5 | 77.06% | 99.64% | 100% | 100% |
| Symbolic-Neural$_{topdown}$ | 7 | 93.03% | 100.00% | 100% | 100% |
| Symbolic-Neural$_{topdown}$ | 9 | 80.76% | 100.00% | 100% | 100% |
| IG$_{bert\_\{mask|200\}}$ | 1 | 72.94% | 99.68% | – | – |
| IG$_{bert\_\{mask|200\}}$ | 3 | 67.15% | 96.91% | – | – |
| IG$_{bert\_\{mask|200\}}$ | 5 | 69.21% | 90.01% | – | – |
| IG$_{bert\_\{mask|200\}}$ | 7 | 57.03% | 86.36% | – | – |
| IG$_{bert\_\{mask|200\}}$ | 9 | 66.20% | 86.29% | – | – |

Figure 6: Top 4 precision and span label precision.

probability of the shortcut span. Please note the precision of shortcut span labels predicted by our model is 100%. Such results demonstrate again that our model is self-interpretable and could reflect the model's rationales by span labels. Samples of label trees with shortcut tokens are shown in Appendix A.5

## 4.3 CONSISTENCY WITH HUMAN RATIONALES

To evaluate the consistency of the span labels learned by our model with human rationales, we design a constituent-level attribution task. Specifically, we hide the gold span positions in NER and slot-filling datasets to see whether our model is able to recover gold spans and labels. So only raw text and sentence-level gold labels are visible to models. We then train models as multi-label classification tasks and evaluate span positions learned unsupervisedly by models.

Figure 7: A sample of our method on semi-supervised slot filling. The ground truths are Denver, Oakland, afternoon, 5 pm, nonstop for each slot correspondingly. However, the last three are reasonable even though different from the ground truths.

**Data set.** We report F1 scores on the following data sets: ATIS (Hakkani-Tur et al., 2016), MITRestaurant (Liu et al., 2013a) and MITMovie (Liu et al., 2013b). ATIS is a slot-filling task and the others are NER tasks.

**Baselines.** We include two baselines with attribution ability on multi-label tasks: integrated-gradient(IG) (Sundararajan et al., 2017) and multi-instance learning (Angelidis & Lapata, 2018). We follow the setup in Sec 4.1 and report the results of the last epoch. For IG, we set the interpolation steps as 200 and use the same BERT in the last section as the encoder, filter the attribution of each token by a threshold and select filtered positions as outputs. We use zero vectors and [MASK] embeddings as the baselines for IG as Bastings et al. (2022) find the latter one could significantly improve its performance. Considering IG scores not having explicit meaning, we allow IG to adjust thresholds according to the test datasets. We report the best results of both baselines and corresponding thresholds. Please find the full version of the table in Appendix A.4. For MIMLL, we select the span with the max attention score for a specified label. Please find details in Appendix A.7.

**Metrics.** We denote the predicted span set as $\mathcal{P}$ and gold span set as $\mathcal{G}$ and the overlap of $\mathcal{P}$ and $\mathcal{G}$ with the same labels as $\mathcal{O}$. Then we have:

$$\text{prec} = \frac{\sum_{o\in\mathcal{O}} o.j - o.i + 1}{\sum_{p\in\mathcal{P}} p.j - p.i + 1}, \text{recall} = \frac{\sum_{o\in\mathcal{O}} o.j - o.i + 1}{\sum_{g\in\mathcal{G}} g.j - g.i + 1}, \text{F1} = \frac{2 * (\text{prec} \cdot \text{recall})}{(\text{prec} + \text{recall})} \quad (7)$$

| Model | Thres. | Slot-filling | | | | Thres. | sls-movie-eng | | | |
|---|---|---|---|---|---|---|---|---|---|---|
| length | | all | $1-2$ | $2-3$ | $3-5$ | | all | $1-2$ | $2-5$ | $>5$ |
| ratio | | 100 | 95.84 | 3.70 | 0.46 | | 100 | 55.60 | 42.75 | 1.65 |
| IG$_{BERT\{mask\|200\}}$ | 0.3 | 50.28 | 51.13 | 37.15 | 15.87 | 0.2 | 57.19 | **60.07** | 55.79 | 34.36 |
| IG$_{BERT\{zero\|200\}}$ | 0.4 | **56.62** | **57.42** | **46.15** | 18.46 | 0.3 | 47.59 | 50.53 | 46.12 | 23.80 |
| MIMLL | N.A. | 11.11 | 10.84 | 17.37 | 16.74 | N.A. | 14.55 | 14.11 | 14.76 | 17.43 |
| Symbolic-Neuron | N.A. | 35.30 | 35.38 | 33.78 | 34.01 | N.A. | 53.04 | 50.61 | 54.99 | 57.77 |
| Symbolic-Neuron$_{exclusive}$ | N.A. | 32.13 | 32.37 | 30.62 | 16.33 | N.A. | 52.89 | 50.45 | 54.55 | **61.15** |
| Symbolic-Neuron$_{topdown}$ | N.A. | 32.86 | 32.91 | 32.06 | 32.88 | N.A. | 53.15 | 51.59 | 54.21 | 59.08 |
| Symbolic-Neuron$_{top./excl.}$ | N.A. | 42.01 | 42.28 | 38.69 | **34.95** | N.A. | **57.82** | 56.54 | **58.87** | 59.82 |
| Model | Thres. | sls-movie-trivial | | | | Thres. | sls-restaurant | | | |
| length | | all | $1-2$ | $2-5$ | $>5$ | | all | $1-2$ | $2-5$ | $>5$ |
| ratio | | 100 | 7.57 | 57.07 | 35.36 | | 100 | 40.87 | 57.89 | 1.24 |
| IG$_{BERT\{mask\|200\}}$ | 0.02 | 47.69 | 38.63 | 45.27 | 50.83 | 0.2 | 50.10 | **51.06** | 50.11 | 37.73 |
| IG$_{BERT\{zero\|200\}}$ | 0.1 | 42.34 | 31.20 | 39.00 | 46.57 | 0.2 | 43.65 | 45.85 | 43.07 | 30.23 |
| MIMLL | N.A. | 61.77 | 42.48 | 52.03 | 69.02 | N.A. | 7.58 | 7.40 | 7.73 | 6.08 |
| Symbolic-Neuron | N.A. | 67.30 | 41.11 | 60.18 | 75.18 | N.A. | 48.07 | 42.93 | 50.89 | 45.60 |
| Symbolic-Neuron$_{exclusive}$ | N.A. | 63.60 | 44.75 | 58.89 | 68.80 | N.A. | 49.46 | 44.28 | 52.22 | **48.20** |
| Symbolic-Neuron$_{topdown}$ | N.A. | 68.55 | 41.62 | 60.74 | 77.07 | N.A. | 47.43 | 43.42 | 49.83 | 40.41 |
| Symbolic-Neuron$_{top./excl.}$ | N.A. | **70.83** | **45.92** | **64.32** | **77.73** | N.A. | **52.52** | 49.14 | **54.67** | 44.27 |

Table 2: F1 scores for semi-supervised slot filling and NER whose golden span positions are hidden. "Thres." is short for threshold.

**Results and discussion.** From Table 2, one observation is that models with the mutual-exclusiveness constraint achieve better F1 scores. Such results illustrate that a stronger inductive bias is more helpful for models to learn constituent-label alignments. Besides, we find the Neural-Symbolic models significantly outperform the MIMLL and IG baselines on the NER datasets but trail the IG on the slot-filling task. Through studying the outputs of our method, with a sample shown in Figure 7, we find that our model tends to recall long spans while the ground truths in ATIS tend to be short spans. We also find that on sls-movie-trivial, MIMLL significantly outperforms IG. So we hypothesize that the distribution of golden span lengths may affect results. We divide sentences into buckets according to the average golden span length and compute F1 scores for each bucket, as shown in Table 2. Interestingly, we find that the scores of IG decline significantly with increasing

span lengths, while our method performs well on all the buckets. In addition, we argue that the F1 scores on the NER datasets can reflect interpretability more objectively, because the boundaries of proper nouns are clear and objective, while the choice of slots is relatively ambiguous about whether to include prepositions, modal verbs, etc.

## 4.4 CASE STUDY & POTENTIAL APPLICATIONS

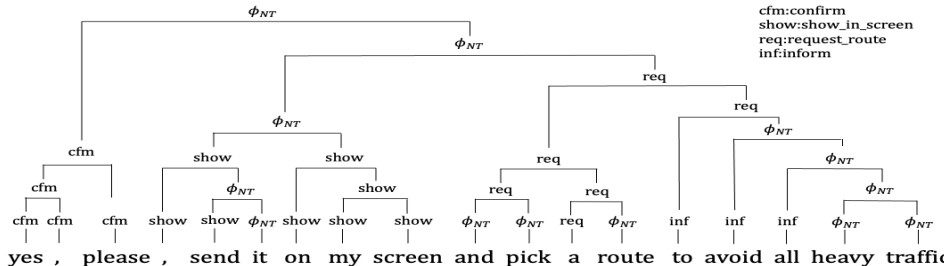

Figure 8: A sample of the symbolic-neural model on Navigator with the top-down encoder.

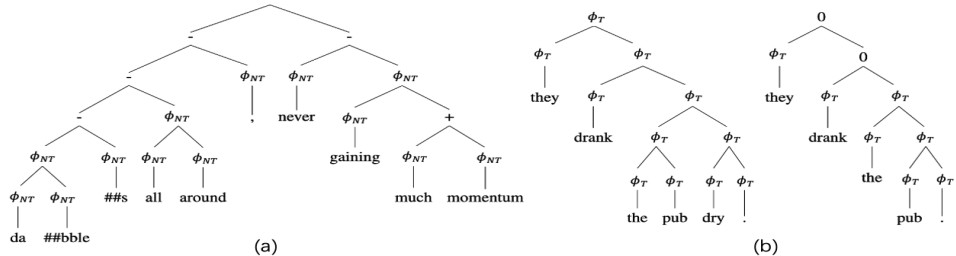

Figure 9: Samples of the symbolic-neural model with the top-down encoder.

We output the label trees generated by our model trained on the Navigator, SST-2, and CoLA to observe whether the model has sufficient interpretability. From Figure 8 we can find our method is able to learn potential alignments of intents and texts and show them explicitly. This can be used in multi-intent NLU systems to help determine the attribution of slots to corresponding intents. We also study the difference between generated label trees of the vanilla Symbolic-Neural model and the Symbolic-Neural$_{topdown}$. The cases could be found in Appendix A.12. We find the vanilla Symbolic-Neural model fails to deal with multi-intent cases. Such an observation verifies the necessity of introducing the top-down encoder. For SST-2, as there are no neutral samples, we randomly sampled sentences from Wiki-103 as neutral texts and force all nodes to be $\phi_{NT}$ by the mean squared error loss. Figure 9(a) shows the sentiment polarity of each constituent and the polarity reversal of "never". Such a characteristic could be used for text mining by gathering the minimal spans of a specified label. We also study the generated label trees on CoLA, a linguistic acceptance data set. We transfer the task to a grammar error detection problem by converting the label "1" to $\phi$ as "1" means no error is found in a sentence. Figure 9(b) shows it's able to detect incomplete constituents and may help in applications like grammar error location. More cases could be found in the Appendix.

## 5 CONCLUSION & LIMITATION

In this paper, we propose a novel label extraction framework based on a simple inductive bias and model single/multi-label text classification in a unified way. We discuss how to build a probabilistic model to maximize the valid potential label trees by leveraging the internal representations of a structured language model as symbolic interfaces. Our experiment results show our method achieves inherent interpretability on various granularities. The generated label trees could have potential values in various unsupervised tasks requiring constituent-level outputs.

Regarding to the limitation of our work, we require that the labels corresponding to the texts in the dataset have a certain degree of diversity, thus forcing the model to learn self-consistent constituent-label alignments. For example, in ATIS, almost all training samples have the same labels like "from-loc.city_name" and "toloc.city_name". That's why our model fails to accurately associate these two labels with correct spans in Figure 7.

## 6 REPRODUCIBILITY STATEMENT

In the supplemental, we include a zip file containing our code and datasets downloading linkage. We've also included in the supplemental the scripts we run all baselines and the Symbolic-Neural models.

## 7 ACKNOWLEDGEMENT

This work was supported by Ant Group through CCF-Ant Research Fund. We thank the Aliyun EFLOPS team for their substantial support in designing and providing a cutting-edge training platform to facilitate fast experimentation in this work. We also thank Jing Zheng for his help in paper revising and code reviewing.

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

# A APPENDIX

## A.1 RELATED WORKS

**Structured language models.** Many attempts have been made to develop structured language models. Pollack (1990) proposed to use RvNN as a recursive architecture to encode text hierarchically, and Socher et al. (2013) showed the effectiveness of RvNNs with gold trees for sentiment analysis. However, both approaches require annotated trees. Gumbel-Tree-LSTMs (Choi et al., 2018) construct trees by recursively selecting two terminal nodes to merge and learning composition probabilities via downstream tasks. CRvNN (Chowdhury & Caragea, 2021) makes the entire process end-to-end differentiable and parallel by introducing a continuous relaxation. However, neither Gumbel-Tree-LSTMs nor CRvNN mention the pretraining mechanism in their work. URNNG (Kim et al., 2019) proposed the first architecture to jointly pretrain a parser and an encoder based on RNNG (Dyer et al., 2016). However, its $O(n^3)$ time and space complexity makes it hard to pretrain on large-scale corpora. ON-LSTM and StructFormer (Shen et al., 2019; 2021) propose a series of methods to integrate structures into LSTM or Transformer by masking information in differentiable ways. As the encoding process is still performed in layer-stacking models, there are no intermediate representations for tree nodes. Maillard et al. (2017) propose an alternative approach, based on a differentiable CKY encoding. The algorithm is differentiable by using a soft-gating approach, which approximates discrete candidate selection by a probabilistic mixture of the constituents available in a given cell of the chart. While their work relies on annotated downstream tasks to learn structures,Drozdov et al. (2019) propose a novel auto-encoder-like pretraining objective based on the inside-outside algorithm Baker (1979); Casacuberta (1994) but is still of cubic complexity. To tackle the $O(n^3)$ limitation of CKY encoding, Hu et al. (2021) propose an MLM-like pretraining objective and a pruning strategy, which reduces the complexity of encoding to linear and makes the model possible to pretrain on large-scale corpora.

**Multi-Instance Learning.** Multi-Instance learning (MIL) deals with problems where labels are associated with groups of instances or bags (spans in our case), while instance labels are unobserved. The goal is either to label bags Keeler et al. (1990); Dietterich et al. (1997); Maron & Ratan (1998) or to simultaneously infer bag and instance labels Zhou et al. (2009); Kotzias et al. (2015). Angelidis & Lapata (2018) apply MIL to segment-level sentiment analysis based on an attention-based scoring method. In our work, we refine instances to different semantic granularities and consider hierarchical relationships between instances.

**Model Interpretability.** In the line of work on model interpretability, many approaches have been proposed. Ribeiro et al. (2016); Lundberg & Lee (2017) try to generate explanation for prediction. Baehrens et al. (2010); Simonyan et al. (2014); Sundararajan et al. (2017) analyze attribution by gradients. The above-mentioned methods are all posthoc. Kim et al. (2020); Cao et al. (2020); Chen & Ji (2020) apply masks on the model input in text classification to obtain token weights, but single-dimensional weights are not enough to reflect multi-label interpretation. Alvarez-Melis & Jaakkola (2018); Rudin (2019) argue interpretability should be an inherent property of a deep neural network and propose corresponding model architectures. However, all the above-mentioned methods are not able to generate constituent-level interpretability.

## A.2 DYNAMIC PROGRAMMING BASED ON FULL PERMUTATION

A naive way to estimate $P(\hat{t}_{i,j}^{[\mathcal{Y}(\mathcal{M})]}|t_{i,j})$ is to enumerate all possible state spaces and sum them up via dynamic programming. We use $X_{i,j}^{\mathcal{M}}$ short for $P(\hat{t}_{i,j}^{[\mathcal{Y}(\mathcal{M})]}|t_{i,j})$. Let $\mathcal{M}_l$ and $\mathcal{M}_r$ denote a pair of sets subject to $\mathcal{M}_l \cup \mathcal{M}_r = \mathcal{M}$. Let $\mathcal{C}(\mathcal{M})$ denote the set containing all valid $\mathcal{M}_l$ and $\mathcal{M}_r$ pairs. Figure 5(a) discusses all potential combinations of $\mathcal{M}_l$ and $\mathcal{M}_r$ when $|\mathcal{M}| > 1$.

If $|\mathcal{M}| > 1$, let $\mathcal{C}(\mathcal{M})$ be the set of all potential pairs where $\mathcal{Y}(\hat{t}_{i,k}) \cup \mathcal{Y}(\hat{t}_{k+1,j}) = \mathcal{M}$.
If $|\mathcal{M}| = 1$, it's similar to the case described in Figure 4.
If $\mathcal{M} = \phi$, $n_{i,j}$ could only be associated with $\phi_T$ or $\phi_{NT}$ with $\mathcal{M}_l = \phi$ and $\mathcal{M}_r = \phi$.

Finally the transition function for $t_{i,j}$ where $i < j$ is:

$$X_{i,j}^{\mathcal{M}} = \begin{cases} P(\phi_{NT}|n_{i,j}) \cdot \sum_{\{\mathcal{M}_l, \mathcal{M}_r\} \in \mathcal{C}(\mathcal{M})} X_{i,k}^{\mathcal{M}_l} X_{k+1,j}^{\mathcal{M}_r} & , |\mathcal{M}| > 1 \\ P(m|n_{i,j}) + P(\phi_{NT}|n_{i,j})(X_{i,k}^{\phi} X_{k+1,j}^{\mathcal{M}} + X_{i,k}^{\mathcal{M}} X_{k+1,j}^{\phi} + X_{i,k}^{\mathcal{M}} X_{k+1,j}^{\mathcal{M}}) & , \mathcal{M} = \{m\} \\ P(\phi_T|n_{i,j}) + P(\phi_{NT}|n_{i,j})(X_{i,k}^{\phi} X_{k+1,j}^{\phi}) & , \mathcal{M} = \phi \end{cases}$$
(8)

When $i = j$, we have:

$$X_{i,j}^{\mathcal{M}} = \begin{cases} 0 & , |\mathcal{M}| > 1 \\ P(m|n_{i,j}) & , \mathcal{M} = \{m\} \\ P(\phi_T|n_{i,j}) + P(\phi_{NT}|n_{i,j}) & , \mathcal{M} = \phi \end{cases}$$
(9)

The transition function works in a bottom-up manner and iterates all possible $\mathcal{M} \subseteq \mathcal{T}$. $X_{1,|\mathbf{S}|}^{\mathcal{T}}$ is the final probability. Even though, iterating $\mathcal{C}(\mathcal{M})$ and all $\mathcal{M} \subseteq \mathcal{T}$ is of exponential complexity, so it only works when $|\mathcal{T}|$ is small.

## A.3 PRETRAIN BERT AND FAST-R2D2 FROM SCRATCH

The dataset WikiBooks originally used to train BERT (Devlin et al., 2019) is a combination of English Wikipedia and BooksCorpus (Zhu et al., 2015). However, BooksCorpus is no longer publicly available. So it's hard to pretrain Fast-R2D2 on the same corpus, making it impossible to compare fairly with the publicly available BERT model. Considering the limited GPU resources, we pretrain both BERT and Fast-R2D2 from scratch on Wiki-103. We train BERT from scratch following the tutorial by Huggingface [5] with the masked rate set to 15%. The vocabulary of BERT and Fast-R2D2 is kept the same as the original BERT. As demonstrated in RoBERTa (Liu et al., 2019) that the NSP task is harmful and longer sentence is helpful to improve performance in downstream tasks, we remove the NSP task and use the original corpus that is not split into sentences as inputs. For Fast-R2D2, WikiText103 is split at the sentence level, and sentences longer than 200 after tokenization are discarded (about 0.04‰ of the original data). BERT is pretrained for 60 epochs with a learning rate of $5 \times 10^{-5}$ and batch size 50 per GPU on 8 A100 GPUs. Fast-R2D2 is pretrained with learning rate of $5 \times 10^{-5}$ for the transformer encoder and $1 \times 10^{-3}$ for the parser. Please note that the batch size of Fast-R2D2 is dynamically adjusted to ensure the total length of sentences in a batch won't exceed a certain maximum threshold, to make the batch size similar to that of BERT, the maximum threshold is set to 1536. Because the average sentence length is around 30 for Wiki103, the average batch size of Fast-R2D2 is around 50 which is similar to that of BERT.

---

[5]https://huggingface.co/blog/how-to-train

## A.4 THE FULL VERSION OF THE SPAN ATTRIBUTION TASK.

| Model | Thres. | Slot-filling | | | | Thres. | sls-movie-eng | | | |
|---|---|---|---|---|---|---|---|---|---|---|
| length | | all | $1-2$ | $2-3$ | $3-5$ | | all | $1-2$ | $2-5$ | $>5$ |
| ratio | | 100 | 95.84 | 3.70 | 0.46 | | 100 | 55.60 | 42.75 | 1.65 |
| $IG_{BERT\{mask|200\}}$ | 0.2 | 47.80 | 48.35 | 40.51 | 17.72 | 0.1 | 51.40 | 51.57 | 51.74 | 42.69 |
| $IG_{BERT\{mask|200\}}$ | 0.3 | 50.28 | 51.13 | 37.15 | 15.87 | 0.2 | 57.19 | 60.07 | 55.79 | 34.36 |
| $IG_{BERT\{mask|200\}}$ | 0.4 | 49.82 | 51.03 | 30.96 | 10.53 | 0.3 | 56.84 | 62.13 | 53.71 | 26.15 |
| $IG_{BERT\{zero|200\}}$ | 0.3 | 53.88 | 54.33 | 48.93 | 23.68 | 0.2 | 45.57 | 46.20 | 45.72 | 31.88 |
| $IG_{BERT\{zero|200\}}$ | 0.4 | 56.62 | 57.42 | 46.15 | 18.46 | 0.3 | 47.59 | 50.53 | 46.12 | 23.80 |
| $IG_{BERT\{zero|200\}}$ | 0.5 | 56.24 | 57.44 | 39.09 | 13.79 | 0.4 | 46.01 | 51.47 | 42.61 | 12.01 |
| MIMLL | N.A. | 11.11 | 10.84 | 17.37 | 16.74 | N.A. | 14.55 | 14.11 | 14.76 | 17.43 |
| Symbolic-Neuron | N.A. | 35.30 | 35.38 | 33.78 | 34.01 | N.A. | 53.04 | 50.61 | 54.99 | 57.77 |
| Symbolic-Neuron$_{exclusive}$ | N.A. | 32.13 | 32.37 | 30.62 | 16.33 | N.A. | 52.89 | 50.45 | 54.55 | **61.15** |
| Symbolic-Neuron$_{topdown}$ | N.A. | 32.86 | 32.91 | 32.06 | 32.88 | N.A. | 53.15 | 51.59 | 54.21 | 59.08 |
| Symbolic-Neuron$_{top./excl.}$ | N.A. | 42.01 | 42.28 | 38.69 | **34.95** | N.A. | **57.82** | **56.54** | **58.87** | 59.82 |
| Model | Thres. | sls-movie-trivial | | | | Thres. | sls-restaurant | | | |
| length | | all | $1-2$ | $2-5$ | $>5$ | | all | $1-2$ | $2-5$ | $>5$ |
| ratio | | 100 | 7.57 | 57.07 | 35.36 | | 100 | 40.87 | 57.89 | 1.24 |
| $IG_{BERT\{mask|200\}}$ | 0.02 | 47.30 | 30.68 | 41.50 | 54.73 | 0.1 | 46.70 | 45.45 | 47.58 | 41.34 |
| $IG_{BERT\{mask|200\}}$ | 0.05 | 47.69 | 38.63 | 45.27 | 50.83 | 0.2 | 50.10 | 51.06 | 50.11 | 37.73 |
| $IG_{BERT\{mask|200\}}$ | 0.1 | 44.35 | 43.91 | 46.00 | 42.91 | 0.3 | 49.04 | **51.67** | 48.46 | 31.33 |
| $IG_{BERT\{zero|200\}}$ | 0.05 | 41.78 | 25.79 | 36.29 | 49.00 | 0.1 | 40.01 | 39.19 | 40.72 | 32.92 |
| $IG_{BERT\{zero|200\}}$ | 0.1 | 42.34 | 31.20 | 39.00 | 46.57 | 0.2 | 43.65 | 45.85 | 43.07 | 30.23 |
| $IG_{BERT\{zero|200\}}$ | 0.2 | 37.26 | 36.49 | 38.01 | 36.68 | 0.3 | 43.07 | 45.45 | 41.10 | 25.95 |
| MIMLL | N.A. | 61.77 | 42.48 | 52.03 | 69.02 | N.A. | 7.58 | 7.40 | 7.73 | 6.08 |
| Symbolic-Neuron | N.A. | 67.30 | 41.11 | 60.18 | 75.18 | N.A. | 48.07 | 42.93 | 50.89 | 45.60 |
| Symbolic-Neuron$_{exclusive}$ | N.A. | 63.60 | 44.75 | 58.89 | 68.80 | N.A. | 49.46 | 44.28 | 52.22 | **48.20** |
| Symbolic-Neuron$_{topdown}$ | N.A. | 68.55 | 41.62 | 60.74 | 77.07 | N.A. | 47.43 | 43.42 | 49.83 | 40.41 |
| Symbolic-Neuron$_{top./excl.}$ | N.A. | **70.83** | **45.92** | **64.32** | **77.73** | N.A. | **52.52** | 49.14 | **54.67** | 44.27 |

## A.5 SAMPLES OF LABEL TREES WITH SHORTCUT TOKENS.

Samples of label trees with shortcut tokens are shown as follows:

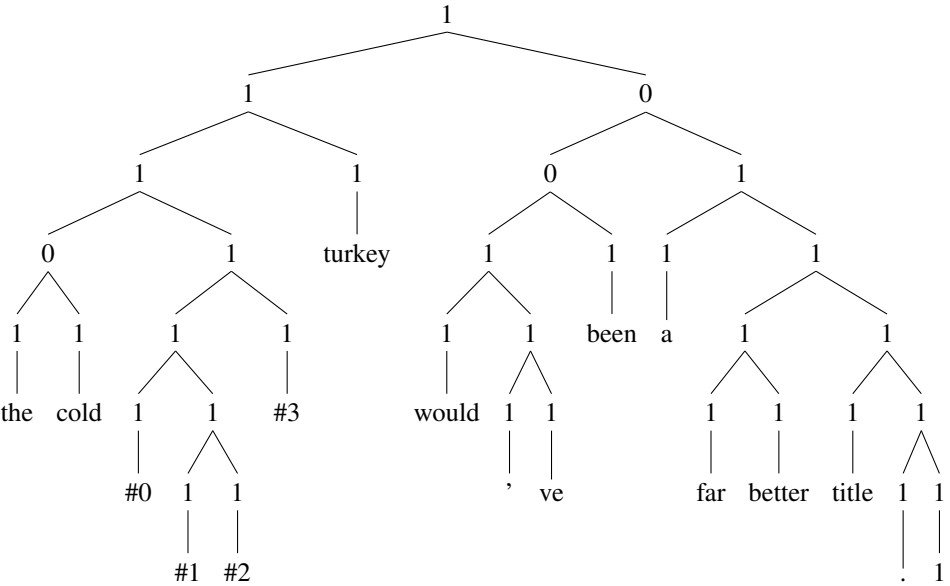

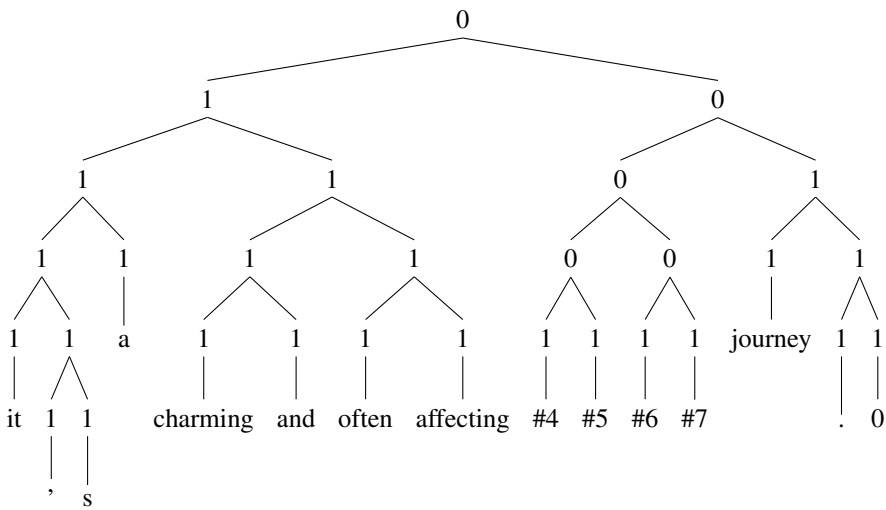

## A.6   MULTI-LABEL LEARNING BASED ON FAST-R2D2

We adopt a canonical multi-instance learning framework used in text classification proposed by Angelidis & Lapata (2018), in which each instance has a representation and all instances are fused by attention. The original work produces hidden vectors $h_i$ for each segment by GRU modules and computes attention weights $a_i$ as the normalized similarity of each $h_i$ with $h_a$.

$$a_i = \frac{exp(\mathrm{h}_i^\mathsf{T}\mathrm{h}_a)}{\sum_i exp(\mathrm{h}_i^\mathsf{T}\mathrm{h}_a)}, p_i = \mathrm{softmax}(W_{cls}h_i + b_{cls}), p_d^{(c)} = \sum_i a_i p_i^{(c)}, \ c \in [1, C]. \tag{10}$$

where $C$ is the total class number, $p_i$ is the individual segment label prediction, $p_d$ is document level predictions. They use the negative log-likelihood of the prediction as an objective function: $L_{cls} = -\sum_d \log p_d^{(y_d)}$. We simply replace segment representations with span representations in our work as the experiment baseline. Specifically, we use the top-down representation $e'_{i,j}$ as the tensor to be attended to and predict the label by $e_{i,j}$:

$$a_{i,j} = \frac{exp(\mathrm{e'}_{i,j}{}^\mathsf{T}\mathrm{h}_a)}{\sum_{m,n\in\mathcal{D}} exp(\mathrm{e'}_{m,n_i}{}^\mathsf{T}\mathrm{h}_a)}, p_{i,j} = \mathrm{softmax}(W_{cls}e_{i,j} + b_{cls}),$$
$$p_d^{(c)} = \sum_{m,n\in\mathcal{D}} a_{m,n}p_{m,n}^{(c)}, \ c \in [1, C]. \tag{11}$$

where $\mathcal{D}$ is the span set for a parsing tree. Please note the MIL model in our baselines is trained together with $\mathcal{L}_{bilm}$ and $\mathcal{L}_{KL}$, whose final loss is $\mathcal{L}_{cls} + \mathcal{L}_{self}$.

## A.7   MULTI-LABEL MULTI-INSTANCE LEARNING BASED ON FAST-R2D2

To support multi-label multi-instance learning, we refactor the above equations to enable them to support attention on different labels. For each label there is vector $h_a^{(c)}$, based on which attention weights $a_i^{(c)}$ is computed as following:

$$a_{i,j}^{(c)} = \frac{exp(\mathrm{e'}_{i,j}^\mathsf{T}\mathrm{h}_a^{(c)})}{\sum_{m,n\in\mathcal{D}} exp(\mathrm{e}_{m,n}^\mathsf{T}\mathrm{h}_a^{(c)})}, p_{i,j}^{(c)} = sigmoid(W_{cls}^{(c)}e_{i,j} + b_{cls}^{(c)}),$$
$$p^{(c)} = \sum_{m,n\in\mathcal{D}} a_{m,n}p_{m,n}^{(c)}, \ c \in [1, C]. \tag{12}$$

The final objective function is $L = -\sum_{c\in\mathcal{T}} \log p^{(c)} - \sum_{c\in\mathcal{F}\backslash\mathcal{T}} \log(1 - p^{(c)})$. In the semi-supervised slot-filling and NER tasks, we let the model predicts labels first and then pick the span with the max attention weight for each label.

## A.8 ABOUT THE CONDITIONAL INDEPENDENCE ASSUMPTION

We argue the independence assumption used in our objective actually is weaker than the one used in conventional multi-label classification tasks. Formally, conventional multi-label classification is the problem of finding a model that maps inputs $\mathbf{x}$ to binary vectors $\mathbf{y}$; that is, it assigns a value of 0 or 1 for each element (label) in $\mathbf{y}$. So the objective of multi-label classification is to minimize: $-\log P(\bigcap_{i\in\mathcal{T}} y_i = 1, \bigcap_{j\in\mathcal{O}} y_j = 0|x)$, where $\mathcal{T}$ denotes the indices for golden labels and $\mathcal{O}$ denotes the indices not in $\mathcal{T}$. It's impossible to tractably estimate it without introducing some conditional independence assumption. By assuming the states of labels are independent of each other, we have:

$$P(\bigcap_{i\in\mathcal{T}} y_i = 1, \bigcap_{j\in\mathcal{O}} y_j = 0|x) \approx P(\bigcap_{i\in\mathcal{T}} y_i = 1|x) \cdot P(\bigcap_{j\in\mathcal{O}} y_j = 0|x) \tag{13}$$

$$\log P(\bigcap_{i\in\mathcal{T}} y_i = 1|x) \approx \log \prod_{i\in\mathcal{T}} P(y_i = 1|x) = \sum_{i\in\mathcal{T}} \log P(y_i = 1|x) \tag{14}$$

$$\log P(\bigcap_{j\in\mathcal{O}} y_j = 0|x) \approx \log \prod_{j\in\mathcal{O}} P(y_j = 0|x) = \sum_{j\in\mathcal{T}} \log P(y_j = 0|x) \tag{15}$$

which could finally be reformulated to the well-known binary cross entropy loss $-\sum_i \hat{y}_i \log y_i + (1 - \hat{y}_i) \log(1 - y_i)$, where $\hat{y}$ is the ground truth and $y$ is the output probability of a model.

The logic of Equation 2 is similar to the above equations. $P(\hat{t}^{[\mathcal{T}\subseteq\mathcal{Y}(\hat{t})]}|t)$ is equivalent to $P(\bigcap_{i\in\mathcal{T}} y_i = 1|x)$ and $P(\hat{t}^{[\mathcal{O}\cap\mathcal{Y}(\hat{t})\neq\phi]}|t)$ is equivalent to $P(\bigcap_{j\in\mathcal{O}} y_j = 0|x)$. But we don't require the independence assumption to estimate the latter.

## A.9 REAL LABEL TREES SAMPLED FROM SYMBOLIC-NEURAL$_{-t/-e}$ ON SST-2

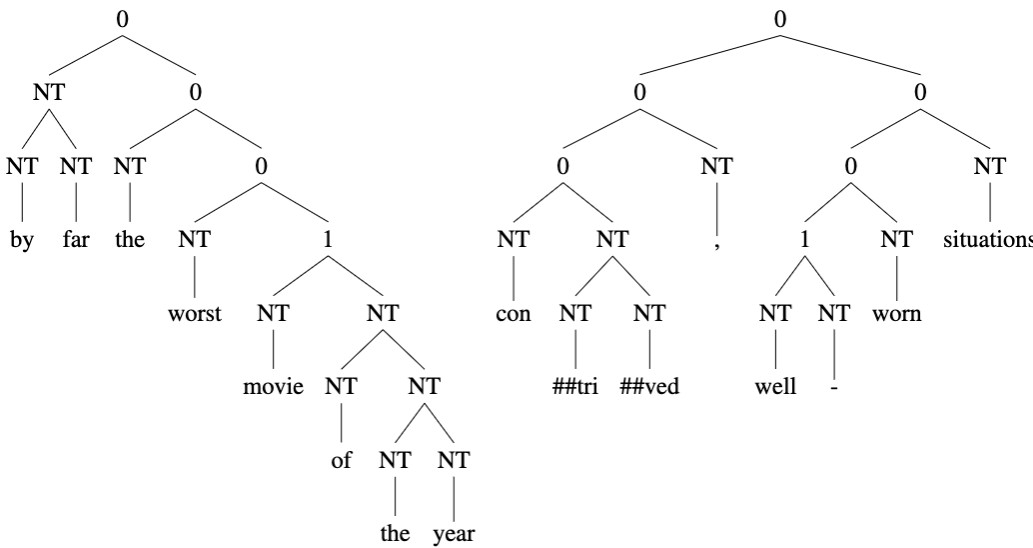

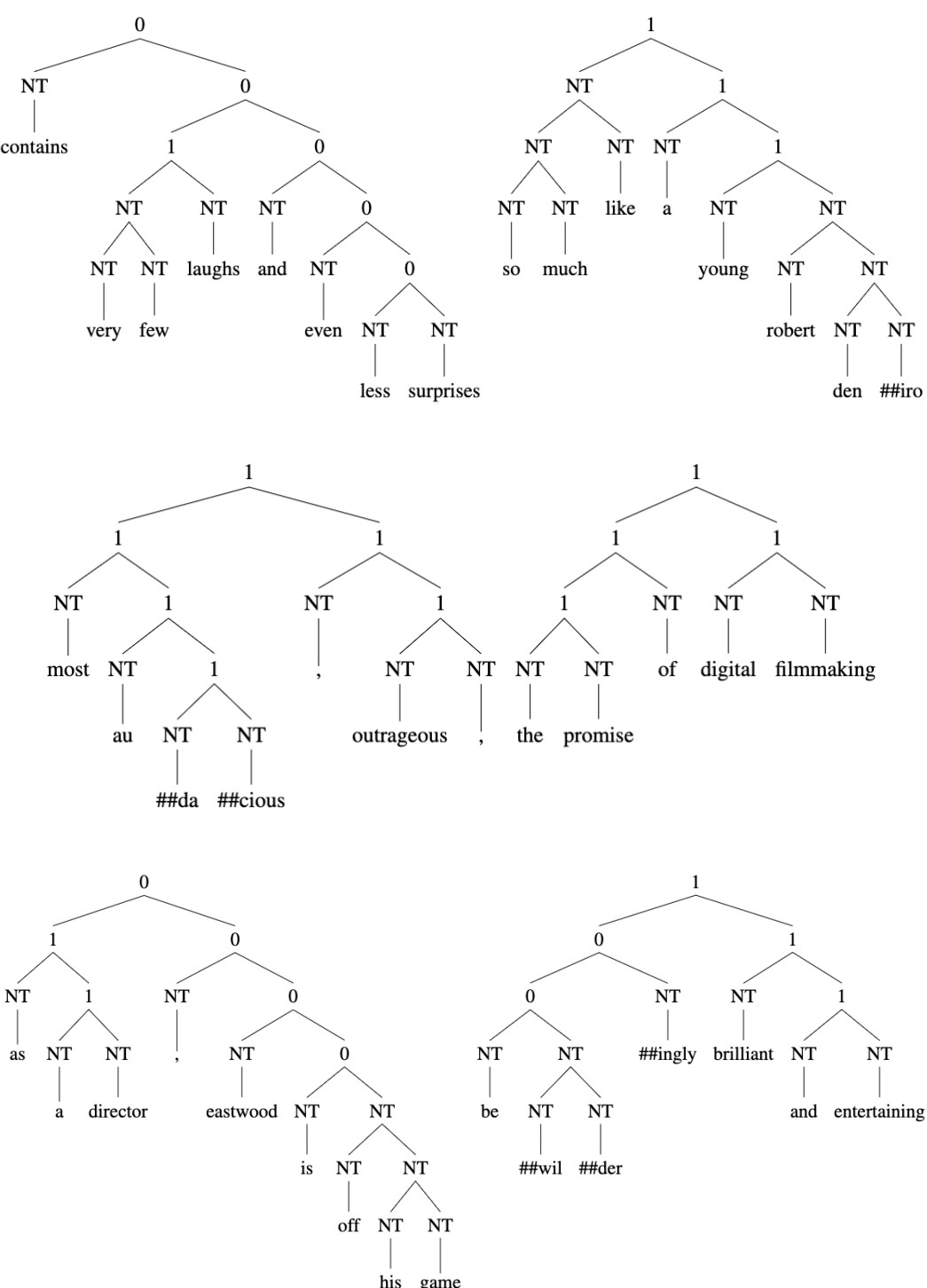

A.10   REAL LABEL TREES SAMPLED FROM SYMBOLIC-NEURAL$_{-t/-e}$ ON SST-2

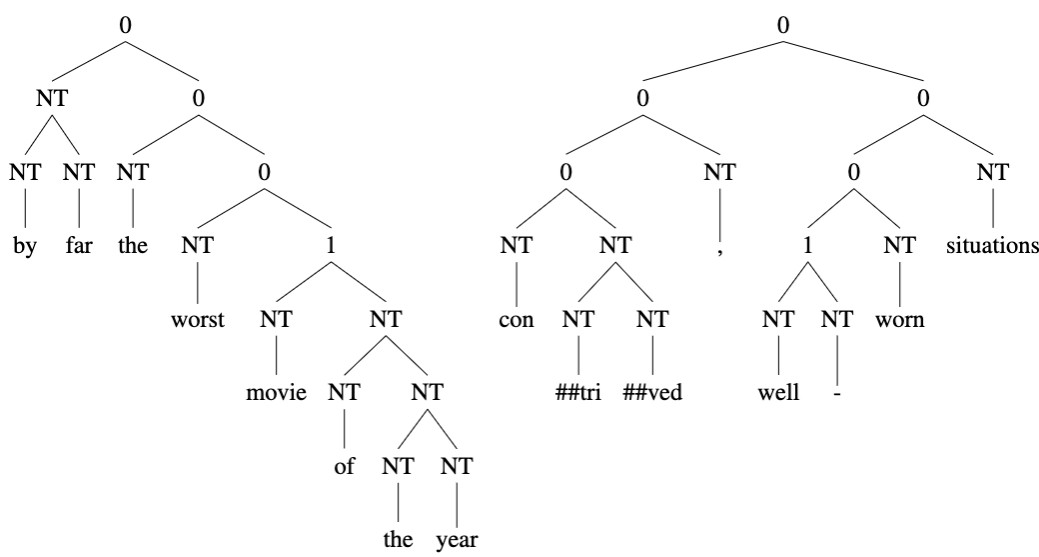

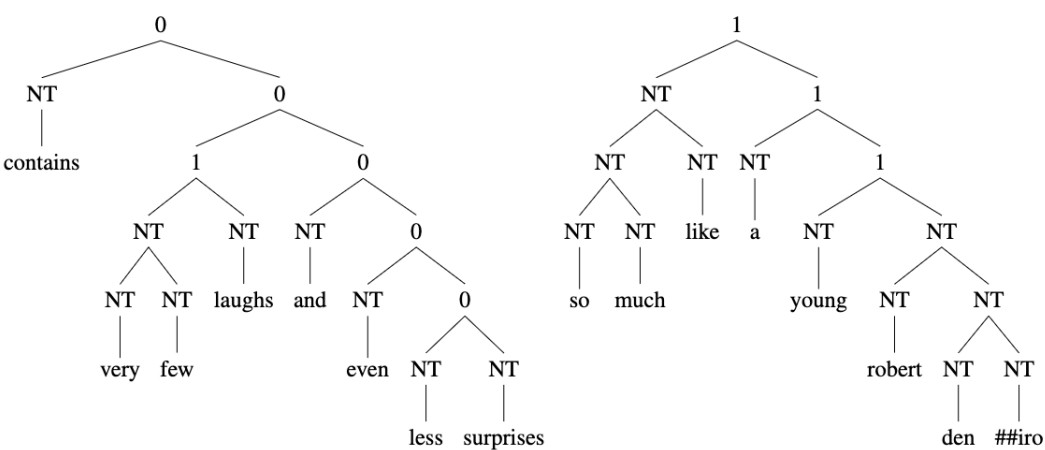

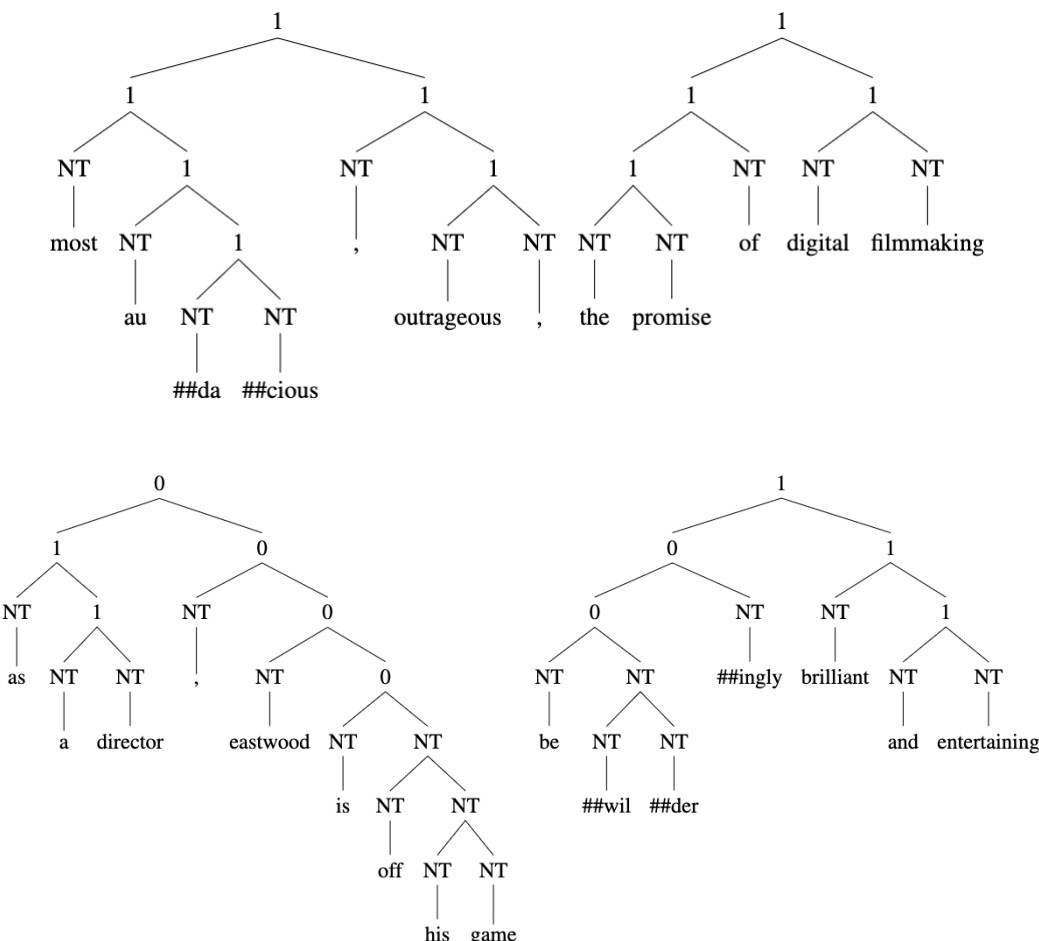

## A.11    REAL LABEL TREES SAMPLED FROM SYMBOLIC-NEURAL$_{-t/-e}$ ON COLA

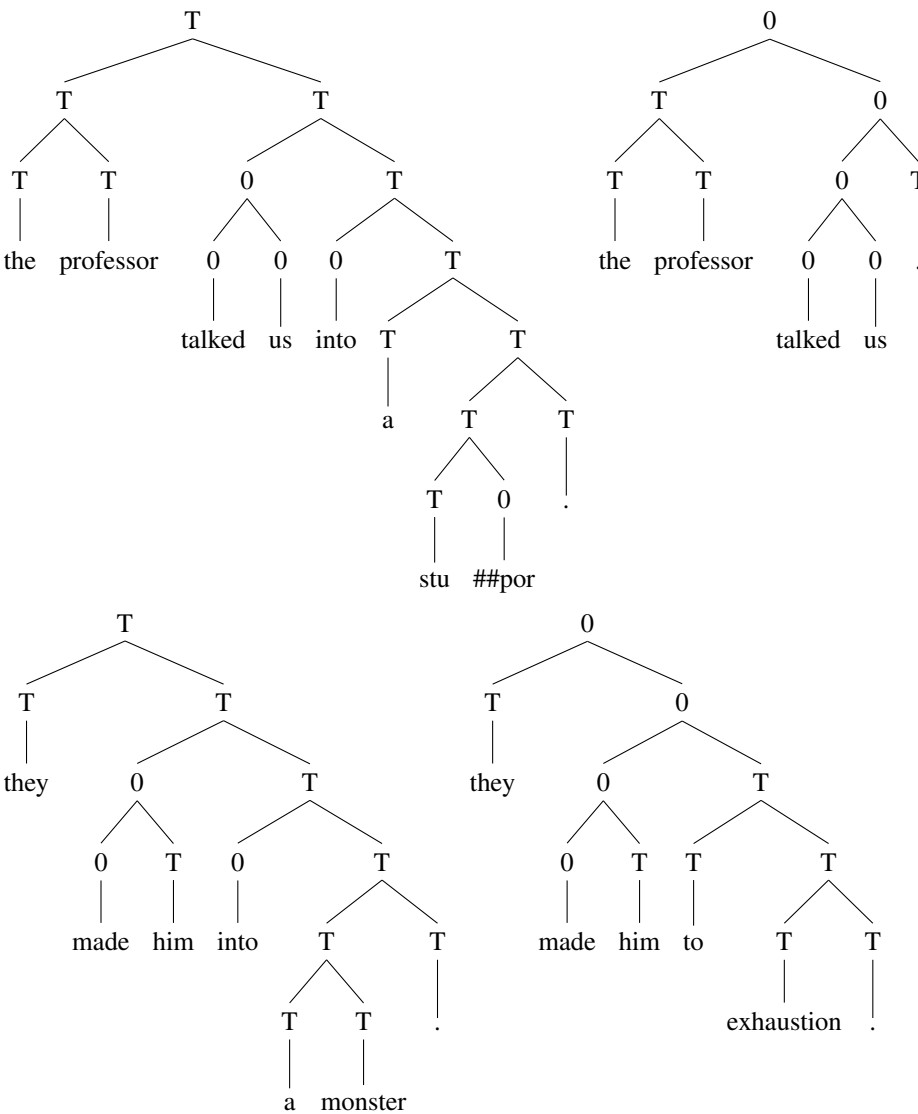

## A.12    SAMPLED LABEL TREES IN ATIS

We sample label trees from Neural-Symbolic$_{-t/-e}$ and Neural-Symbolic$_{+t/-e}$ respectively for observation. Ground truths are annotated in brackets.

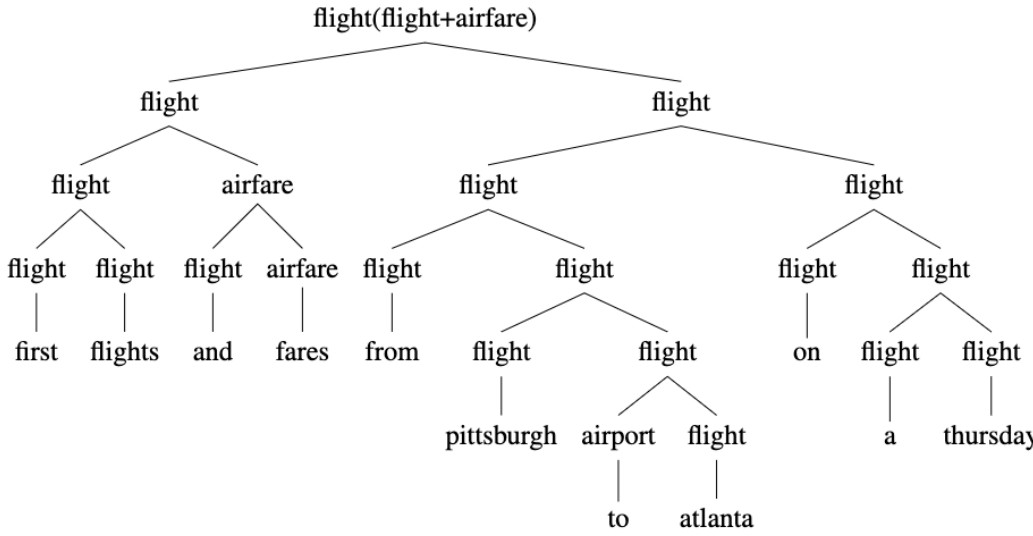

Figure 10: The label tree generated by Neural-Symbolic w/o the topdown encoder.

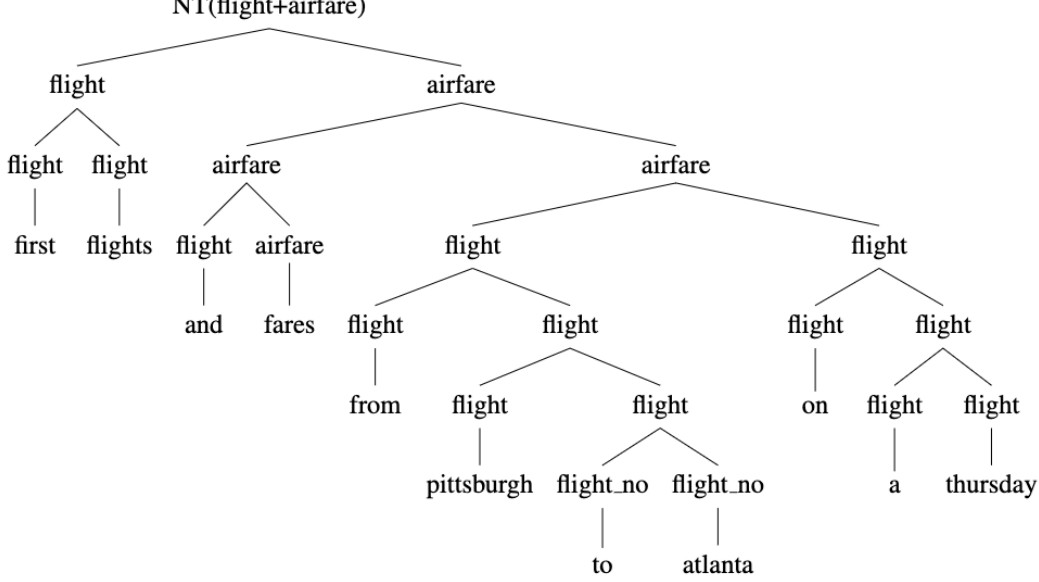

Figure 11: The label tree generated by Neural-Symbolic$_{topdown}$

## A.13    SAMPLED LABEL TREES IN NAVIGATOR

2:request_route, 3:appreciate, 4:request_address, 6:navigate

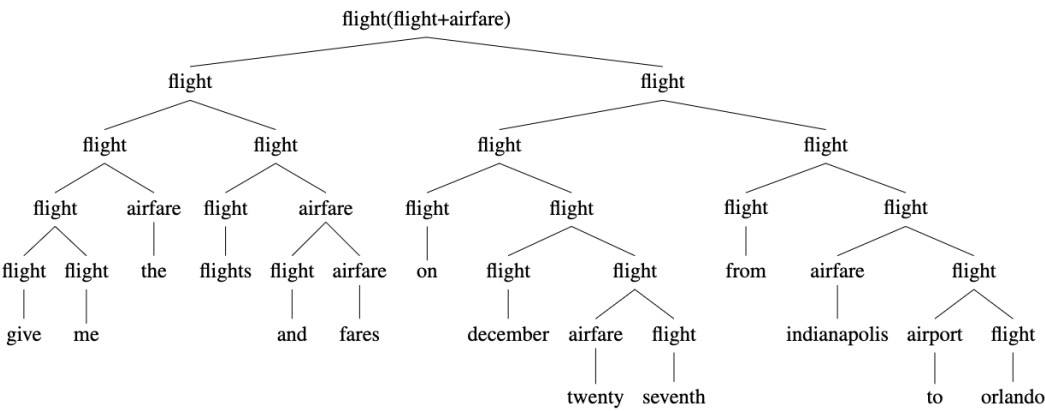

Figure 12: The label tree generated by Neural-Symbolic

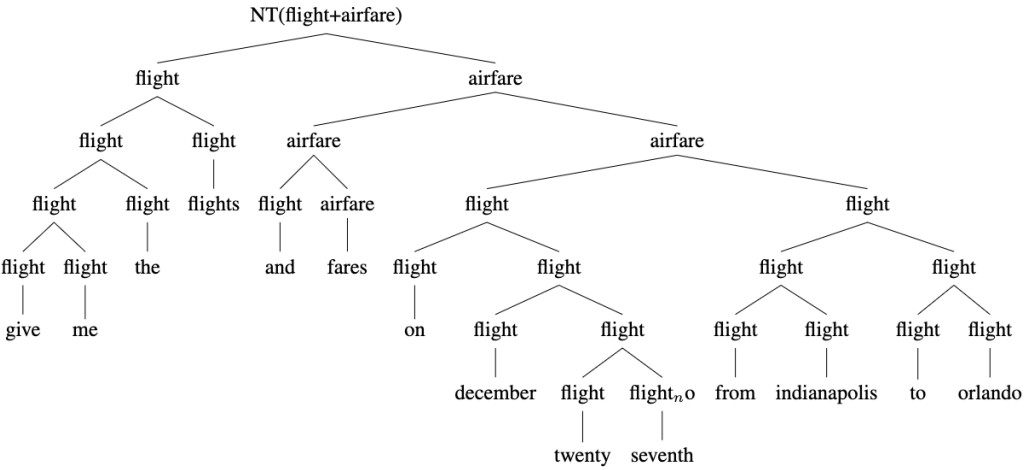

Figure 13: The label tree generated by Neural-Symbolic$_{topdown}$

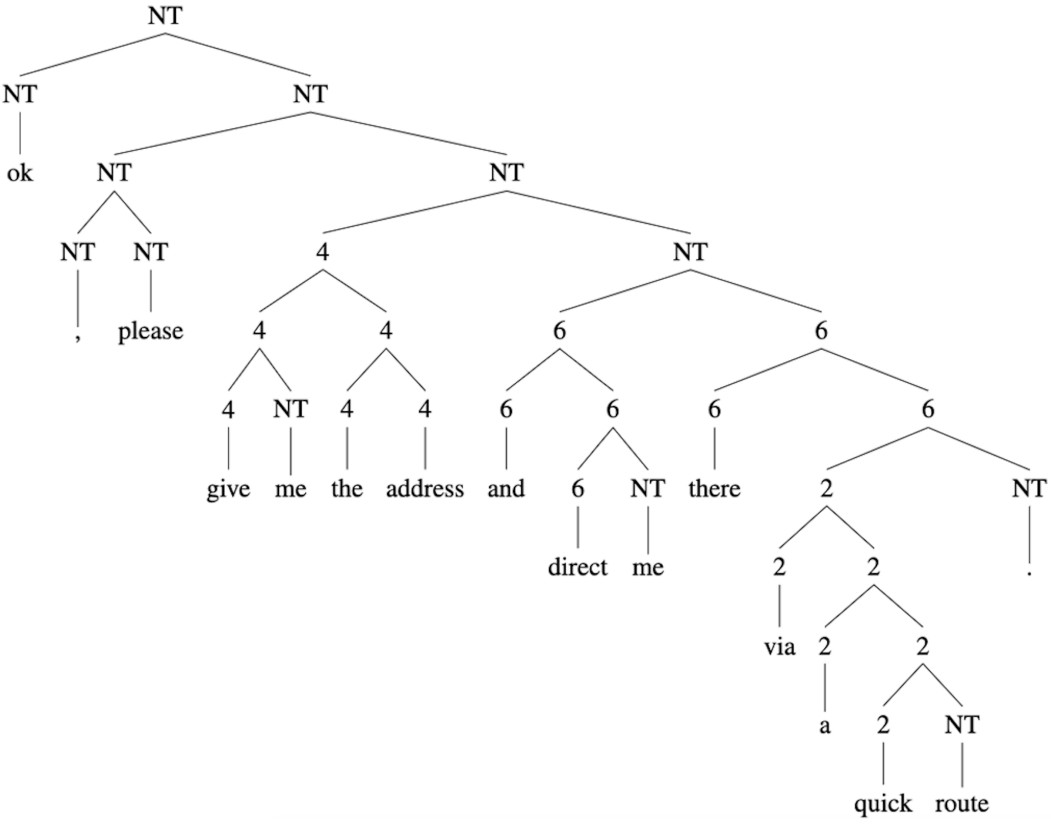

Figure 14: The label tree generated by Neural-Symbolic$_{topdown}$

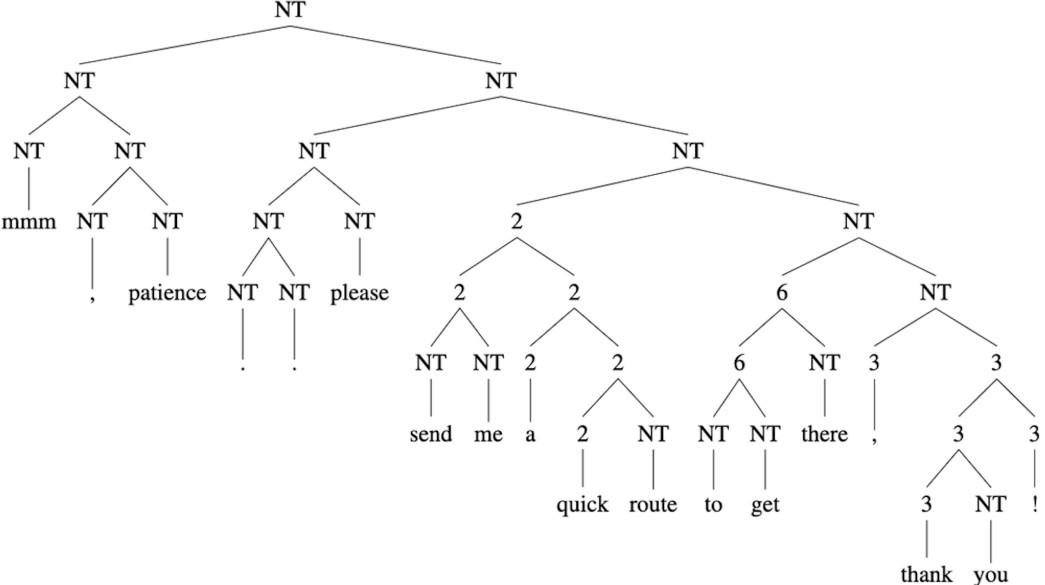

Figure 15: The label tree generated by Neural-Symbolic$_{topdown}$

