# OpenReview forum: "A Multi-Grained Self-Interpretable Symbolic-Neural Model For Single/Multi-Labeled Text Classification"
_ICLR.cc/2023/Conference — ICLR 2023 poster_

### Official Review · Reviewer_AUue · 2022-10-22

**Confidence:** 3
**Correctness:** 3
**Technical Novelty And Significance:** 3
**Empirical Novelty And Significance:** 2
**Recommendation:** 5

**Clarity, Quality, Novelty And Reproducibility:**

[REVISED]

Upon revision, I've come to appreciate the novelty in problem formulation and model/loss formulation of the submission. However, the current lack of clarity of the paper makes it difficult to appreciate these contributions.

[OLD REVIEW]

I find the details in Section 3.2 not useful. It's better to state the algorithm clearly and move the details to an appendix.

**Strength And Weaknesses:**

[REVISED]

STRENGTHS
- The proposed problem is relatively unexplored and interesting.
- The proposed algorithm for computing the label-consistent loss conditioned on a MAP tree is technically novel.
- There is a clear practical benefit. The model works as well as conventional classification models but also induces labeled spans.

WEAKNESSES
- The paper is written *very* confusingly, which is the reason I (and the reviewer Vws4) found it almost impossible to judge the quality of the submission. It's not just one problem but several, which makes the reader wonder
   - What is the model? What parameters are being optimized? The model is never precisely defined but kind of sketched on top of Fast-R2D2. The reader doesn't necessarily know about Fast-R2D2 and has to wonder if this specific to Fast-R2D2 or similar models.
   - What is the new loss? Again despite many equations that technically define the loss, the objective is simply defined in $P(\hat{t}^{[\mathcal{Y}(\hat{t})=\mathcal{T}]} |t)$ and I found the current explanation both verbose and confusing. My main feedback is to really transform the model and training section. Precisely define the model and the objective. The detailed derivation should follow only after the model/objective is crystal clear. I think I've parsed the paper more correctly at this point, but this should never be the burden of the reader.
- I stand by my one previous weakness, which is: the case for the usefulness of label trees needs to be made with more systematic studies instead of qualitative analyses only.


[OLD REVIEW]

STRENGTHS
- The important problem of developing interpretable neuro-symbolic models is considered.

WEAKNESSES
- The method is somewhat straightforward and relies on an existing structured language model.
- The performance results are underwhelming. Table 1 shows that the method attains little or very slight gains over the Fast-R2D2 baselines.
- The case for the usefulness of label trees needs to be made with more systematic studies instead of qualitative analyses only, it seems.

**Summary Of The Paper:**

[REVISED]

Task: The paper considers the relatively unexplored task of making multi-label text classification interpretable by learning a model that induces implicit labeled spans given only sentence-level labels as supervision.

Model: The paper assumes a "backbone" model for unsupervised parsing that gives an embedding $e_{i,j}$ for each span $(i,j)$. A conditional label distribution $P(l|n_{i,j})$ is defined at every considered node $n_{i,j}$ based on $e_{i,j}$. In experiments, the embeddings are augmented to incorporate top-down information.

Training: The model is trained by optimizing $L + L^t$ where $L$ is the usual loss for the backbone model and $L^t$ is the new loss. The new loss fixes the predicted tree from the backbone model $t$ (which changes dynamically during training), then computes the likelihood of $t$ yielding labels that are consistent with the ground-truth labels. Naively, this requires summing over all possible label configurations of $t$ which is exponential in the label size. The paper makes conditional independence assumptions to derive a dynamic programming to approximate this loss.

Experiments: The paper considers multiple single-/multi-label text classification datasets (SST-2, ATIS, etc.). The paper shows that (1) the proposed model works as well as conventional models and (2) the model recovers gold spans and labels in slot-filling/NER tasks.

[OLD REVIEW]

The paper proposes using a structured language model for interpretable classification. The method is based on Fast-R2D2. It proposes a variant of the inside algorithm to compute marginal probabilities of subtrees with gold labels and adds the resulting classification loss to the original objective of Fast-R2D2. Experiments on single- and multi-label classification datasets show that the method is competitive in performance but can yield meaningful trees.

**Summary Of The Review:**

[REVISED]

The paper considers a relatively unexplored problem of interpretable text classification and a new model and training scheme. However, the paper is quite vague and confusing in communicating the model and the training scheme, which makes it very difficult to appreciate these contributions. Even after revising and appreciating the submission much more than before, I think the paper needs substantial restructuring before publication.


[OLD REVIEW]

The paper proposes using a structured language model for interpretable classification, but the method is not surprising and relies on existing structured LMs.

---

> ### Author Response · Authors · 2022-11-18
> **Sincerely hope you can kindly give us more feedback**
>
> Thanks for your review.
>
> We want to clarify that the gold labels are not span-level labels. They are sentence-level labels like \{negative\},\{positive\},\{intent1, intent2\}. Our model can learn to predict span-level labels by just training on the datasets composed of raw text and sentence-level labels like CoLA, SNIPS, etc.
>
> Section 3.2 is supposed to be the core of the paper and we believe it is important to include all the details for deriving and specifying our algorithm. We have rewritten the section for better organization and clarity. We also clarify the logic of the section below to facilitate your understanding. Though tree structures can be predicted by an existing structured language model, the difficulty here is how to associate each node with a single label without span-level gold labels, especially for multi-label cases. So in section 3.2, we introduce an approach that maximizes the total probability of potential trees whose extracted labels are consistent with sentence-level gold labels. The yield algorithm specifies the rule to extract labels. The total probability of those potential trees is computed via dynamic programming based on the symbolic interfaces provided by Fast-R2D2. We elaborate the process in Figure 3 in our revised version. Please note it's not a variant of the inside algorithm because the parsing tree is given.
>
> Regarding the experimental results in Table 1: The goal of the experiment is to show that our model, which is self-interpretable, has competitive performance compared with the baselines, with same training inputs and same pretraining corpus. In other words, we demonstrate that we do not sacrifice downstream task performance for interpretability. We believe it's nontrivial for a self-interpretable model to achieve such results.
>
> We do not see why our method is deemed straightforward. First, note that our model learns to predict span-level labels without access to span-level gold labels during learning, which is a nontrivial and very challenging task. Second, though our method is based on classical symbolic probabilities, it still needs some insight to propose an inductive bias to constrain the model, select a backbone that could provide symbolic interfaces, and design a dynamic programming algorithm on the basis of the first two to tractably maximize the objective function. For the dynamic programming algorithm (section 3.2), note that the most straightforward way to compute the training objective with dynamic programming is intractable, as shown in Appendix A1 of the revised paper. So we have to make additional independence assumptions and decompose the objective function into tractable components. We then design different dynamic programming algorithms to solve different components. In sum, we do not think our method is that straightforward. Could you please give us more details about your understanding and opinions of our method?
>
> Thanks for your affirmation of the importance of developing interpretable neural-symbolic methods.
> Sincerely hope you can kindly give us more feedback.

---

> > ### Comment · Reviewer_AUue · 2022-11-27
> > **Thank you for the response**
> >
> > I apologize for the previous low-quality review. I've revised the review and raised the score to 5. (I've left the old reviews unchanged for record.)

---

> > > ### Author Response · Authors · 2022-11-29
> > > **Thanks for your update**
> > >
> > > Regarding the usefulness of label trees, we conducted two experiments in the revised version. One is the unsupervised label extraction task in Sec4.2 and the other is the shortcut-finding task in Appendix A.4. The former results demonstrate the label trees learned by our model are consistent with human rationales to an extent. The latter shows the learned label trees could reflect the model's rationales. We suppose these experiments are designed to show the usefulness of label trees. If you have further suggestions about systematically studying the usefulness of label trees, could you please let us know?
> > >
> > > In addition, we attached many label trees yielded by our model on different tasks for case studies in the appendix.
> > >
> > > Thanks for your help in improving our paper.

---

### Official Review · Reviewer_1uUN · 2022-10-25

**Confidence:** 3
**Correctness:** 4
**Technical Novelty And Significance:** 3
**Empirical Novelty And Significance:** 4
**Recommendation:** 6

**Clarity, Quality, Novelty And Reproducibility:**

Novelty, Quality and Reproducibility ( code provided ) are solid.  Clarity and describing motivation, for people less knowledgable of the prior work ( particularly in regard to structured LMs and Symbolic Neural modeling ) could be improved.

**Strength And Weaknesses:**

**Strengths**
The empirical results of their method are impressive and pretty thorough and the analysis of explainability from a quantitive standpoint is also interesting.

**Weaknesses**
This paper is too dense and not self contained ( ie, references to charts, chart encoders for example, etc ).
required me going to other cited papers to be clear on certain aspects of the paper.

Section 3.2 and in particular page 4 are very dense and hard to follow initially as is and really could be improved by having a table for variables and a more fully guided example ( another figure in the style of Fig 2 )

Additionally, I think conclusion does a better job of motivating and describing the paper than the intro does, so you may want to take parts of that to help guide the reader, in particular when it comes to understanding when supervision is needed vs not.

**Summary Of The Paper:**

In this paper, the authors propose a Symbolic-Neural (SN) interpretable model for text classification that uses a structured language model Fast-R2D2 as the backbone and during training, maximizes the probability summation of all potential trees whose extracted labels are consistent with a gold label set via dynamic programming with linear complexity.  Given a sentence and parse tree, the model generates a labeled tree which explain constituent spans in the sentence.  The authors conduct experiments on 6 datasets and compare different representation architectures ( uninterpretable dense Sentence Representation and their interpretable Symbolic Neural constituent representation ) showing their method matches or surpasses dense non-interpretable methods.  Additionally, they conduct a quantitative analysis comparing the interpretability of SN to Integrated Gradients and Multi-instance learning by designing a constituent-level attribution task to see how close the results learned by the models in an unsupervised fashion are to human-annotated results.

**Summary Of The Review:**

The empirical results of their method are impressive and thorough as is the analysis of explainability from a quantitive standpoint. My biggest qualms is with clarity of writing and keeping the paper self contained.

---

> ### Author Response · Authors · 2022-11-18
> **Modifications in the revised version.**
>
> Thanks for your review.
>
> We really appreciate your pertinent and constructive feedback and your willingness to read cited papers to understand our paper.
>
> Due to the page limit, it's difficult for us to go deep into the details of R2D2 pretraining. To make it easier for people less knowledgeable of the prior work to understand, we removed the details about the chart-based encoder and pertaining of Fast-R2D2 which is unrelated to the main approach discussed in this paper, and mainly focused on introducing what a structured language model could do, especially its inputs and outputs.
>
> To make Sec 3.2 easier to follow, we moved the dynamic programming algorithm with exponential complexity to the appendix and focused on the dynamic programming algorithm with linear complexity. Following your suggestions, we added Figure 3 as a guided example in the revised version. To facilitate understanding of variables, we used $\hat{t}^{[\mathcal{C}]}$ to denote $\hat{t}$ satisfying condition $\mathcal{C}$ and declared most variables at the beginning of Sec 3.2.
>
> We also rewrote part of the introduction and abstract to describe our motivation more clearly.
>
> Thanks for your help in improving our paper. If there is anything else that can be improved, we would appreciate you telling us.

---

### Official Review · Reviewer_Vws4 · 2022-11-04

**Confidence:** 3
**Correctness:** 2
**Technical Novelty And Significance:** 3
**Empirical Novelty And Significance:** 3
**Recommendation:** 6

**Clarity, Quality, Novelty And Reproducibility:**

Clarity:
- I found the paper very difficult to understand. Much of it is probably my own unfamiliarity with structured language modeling as a subfield. However, there are a number of strange turns of phrase (like "layer stacking model") and areas that could use clearer explanations. Particular points of confusion for me included:
    * The description of the pretraining process for Fast-R2D2. It contained pieces of jargon I was unfamiliar with, such as "chart-based en-coder" and generally seemed too short to fully describe the pretraining. Maybe cut it entirely and just reference the original paper?
    * The description of the interpretability metrics, which I had to read several times before I felt like I understood it. This made me particularly concerned about the clarity of the rest of the paper because interpretability is one of my focus areas. It pushed me more towards thinking that my difficulty understanding other parts of the paper stems from genuine clarity issues, rather than my own unfamiliarity.
    * The description of the training objective (see summary)
    * The yield function (see below)

- There are a number of times where the text detours into fairly technical points without providing context. E.g., when discussing the yield function, there's a detour "For simplicity, we don’t discuss nesting cases in this paper, so there is only one unique non-terminal label...". There's no indication of what a "nesting case" is, why we might encounter one, or why they'd lead to multiple non-terminal labels.

Novelty:
- The primary novel contribution of this work seems to be the optimization objective the authors use to teach structured language models to perform classification at multiple levels of the parse tree. Unfortunately, I couldn't understand how the optimization objective is supposed to work, so I can't really evaluate its significance. I *think*, but am not sure, that the paper's optimization objective allows for classifiers to be trained to predict span level labels without requiring access to span level gold labels. If that's true, then it would be quite the interesting contribution.


Reproducibility:
- Provides the code used for training models / running experiments, which is good.
- However, that code is pretty poorly documented. E.g., no list of library requirements.
- I also do not understand why the authors trained their own version of the Fast-R2D2 model. A public version is available here (https://github.com/alipay/StructuredLM_RTDT). Training a new model seems to just make reproducing this work more difficult. Perhaps the public model was not available at the time of the paper's writing?

**Strength And Weaknesses:**

Strengths:
- Focuses on interpretability from the start, rather than slapping some ad-hoc method on at the end.
- Provides class predictions at each node of the syntax tree. You can see the overall prediction evolving as you walk up the tree, which is neat.
- I appreciate the authors' effort to provide a fair comparison of their approach by training their own BERT model trained with comparable resources.

Weaknesses:
- The authors do not go into much detail regarding the training process of their BERT model, and they do not include the standard pretrained BERT among their comparisons. This makes it difficult to judge the performance of the method relative to the alternatives that are available to a potential user. It would also have been nice to see comparisons of the amount of compute required to train a Fast-R2D2 model as compared to a BERT model.
- I think (though am not entirely sure) that the authors evaluate their method's interpretability by comparing the degree to which highly attributed words in a classified input overlap with human rationales for that input. I think this way of evaluating a method's interpretability is very questionable for two reasons:
    1. Interpretability methods should highlight a model's true basis for making a classification. That basis might be very different from human rationales. The degree of rationale overlap depends on *both* the degree to which the model's rationales overlap with human rationale and the degree to which the highlighted text represents the model's true rationale.
    2. The quality of baselines for rationale overlap metrics is quite erratic, and can vary between different tasks, datasets and models. Integrated gradients and attention scores may be poorly suited to the task at hand. In particular, attention scores can do very poorly as explanation unless scaled by the magnitude of the associated value vectors. See (https://arxiv.org/abs/2004.10102). Failing to apply this correction leads to attention-based attribution scores seeming to highly weight CLS and punctuation tokens, which may be degrading the performance of the author's attention-based baselines. Additionally, the authors do not mention what they used as the baseline for integrated gradients, which (https://arxiv.org/abs/2111.07367) find can have a significant influence on the method's performance.

    - I'd suggest the authors consider another approach to demonstrating the interpretability advantage of their method. E.g., (https://arxiv.org/abs/2111.07367) "poison" a classification dataset with synthetic shortcuts, train classifiers on the poisoned data, and then test if a given interpretability method can pick up on the shortcut.

    - If the above approach represents too much of a burden, an alternative would be to include a wider range of attribution baselines such as:
        - Integrated gradients (preferably with the [MASK] token as the baseline)
        - Attention (while scaling by the attention value vector)
        - L2 norm of the gradients
        - Sum( gradient x input embeddings )
        - ||Max( gradient x input embeddings, 0 )||, a method called NormGrad selective. See (https://arxiv.org/abs/2004.02866)
    - Note that the above alternative only addresses the weakness of using integrated gradients / attention as baselines, not the deeper conceptual issue of rationale overlap being a poor way to evaluate interpretability, regardless of the baselines used for comparison.
- The method seems to require a class of language model different from those currently in common use. This requires potential users to either train their own structured language model (expensive) or to use a publicly released structured language model, of which there seem to be very few.
    - It might be helpful if the authors could demonstrate that their finetuning method works on a publicly released structured language model, which is what potential users would realistically be starting from themselves.

**Summary Of The Paper:**

This paper describes a method of training a classifier based on a pretrained syntax tree extractor called a structured LM. The structured LM produces contextualized representations of each node through a learned composition function applied to the node's children. They then apply an MLP and softmax to produce label probabilities for each node. The authors also define a yield function to gather task-relevant node labels based on the tree structure and node labeling.

However, I don't understand how the optimization objective is supposed to work. The authors mention Tau as representing a "gold label set" associated with a single example, suggesting Tau is supposed to be a set of gold labels on the level of individual nodes / words. However, the authors apply their method to datasets that (to my knowledge) don't have such fine-grained labels, such as CoLA.

The authors say the objective is to "maximize the probability summation of {t_hat | yield(t_hat) = Tau}". I do not understand what this means. What's the probability we're summing up, and how is it computed? Later text mentions how to compute per-node label probabilities, but those seem like different things than what's being summed here. In fact, I don't even understand how there's anything here to sum over. It seems like t_hat is a specific set of possible labels for a specific parse tree, and Tau is either another set of labels for that specific parse tree, or a single label for the entire sentence. Either way, I don't see how there's a space to sum over.

What does it mean for t_hat to be conditioned on yield(t_hat) = Tau? I'd assume it means that we're restricting the space from which we're drawing t_hat to only include elements whose yield is equal to Tau. However, the previous text just called t_hat one of the possible label trees associated with the best syntax tree of a sentence, with no indication that t_hat is at all restricted.

**Summary Of The Review:**

I tentatively recommend rejecting this paper. My top issues with this paper are its lack of clarity (particularly regarding the optimization objective, which is its primary contribution) and the weakness of its experimental protocol.

Lack of clarity:
- I was unable to follow key aspects of the paper.
- There seem to be many opportunities to improve the explanation of the paper's technical contribution, as I highlighted in the summary section.

Experimental weakness:
- I described my concerns with the interpretability experiment in the weaknesses section.
- I am concerned about the lack of detail provided in the training process of the BERT baseline, and the lack of comparison to the publicly available BERT model, as mentioned in the weaknesses section.

---

> ### Author Response · Authors · 2022-11-18
> **Replies to the concerns**
>
> Thanks for your review.
>
>
> Your detailed feedback is helpful for us to know which part is not clear and thanks for the conceptual correction on model interpretability.
>
> Q1: ``I think, but am not sure, that the paper's optimization objective allows for classifiers to be trained to predict span level labels without requiring access to span level gold labels. If that's true, then it would be quite the interesting contribution."
>
> Yes, that's exactly what we do. Thanks for your highly concise summary. Please allow us to use it in the abstract.
>
> Q2: ``how the optimization objective is supposed to work".
>
> Sorry for the confusion. $\mathcal{T}$ is supposed to be a set of sentence-level labels like \{positive\},\{negative\}, \{intent1, intent2\}. $\hat{t}$ is a random variable representing a label tree transferred from a parsing tree $t$ by associating each node with a label. Each node in the parsing tree has a probability distribution of labels, and the probability of $t$ transferring to $\hat{t}$ is the product of per-node label probabilities. There could be various label trees transferred from a given parsing tree, but only a subset of them could yield labels consistent with $\mathcal{T}$. The total probability of these label trees is what we would like to maximize in training. As it is intractable to exhaust all potential label trees, we propose a dynamic programming algorithm to estimate their probability summation, which is the core idea of Section 3.2. Those per-node label probabilities could be used to build transition functions for the dynamic programming algorithm.
> We have rewritten section 3.2 to clarify the notations and derivations, and added figure 3 to explain the dynamic programming algorithm.
>
> Q3: ``I also do not understand why the authors trained their own version ...Perhaps the public model was not available at the time of the paper's writing".
>
> Yes, when we wrote the paper, the public version was not released yet. Our Fast-R2D2 is pretrained following the same procedure and hyperparameters as the publicly released one. Our model could also be tuned directly based on the public version.
>
> Q4: ``I am concerned about the lack of detail provided in the training process of the BERT baseline, and the lack of comparison to the publicly available BERT model..."
>
> In the revised version, we provide the details for pretraining BERT on WikiText-103 in Appendix 4.2. One reason we did not compare our model to the publicly available BERT model is that BooksCorpus used in BERT pretraining is not publicly available anymore. Another reason is our limited GPU resources. Considering that the backbone we used is a completely different architecture, we pretrain both models from scratch on WikiText-103, which is a very large corpus.
>
> Q5: the baseline for IG
>
> The original baseline used for IG was a zero vector. By replacing it with [MASK] vectors, the performance did change. We record the results of both baselines and update the table in our revised version.
>
> Q6: About jargon.
>
> For the "chart-based encoder", we remove it entirely as suggested.
>
> Q7: About the nesting case.
>
> We added an example in the footnote.
>
> Q8: concerns with the interpretability experiment.
>
> We totally agree with your comments and have revised our paper as follows.
> First, we have corrected our description of the experiment in section 4.2. It is designed to evaluate the consistency of label attribution learned by our model with human rationales. Being able to unsupervisedly learn span labels that are consistent with human rationales has considerable application value. We are aware that IG may be poorly suited to the task, so we allow it to cheat by adjusting the threshold according to the test set.
>
> Second, we have clarified that our model is not an interpretability method but a self-interpretable classification model. Our model's rationales are reflected on the class labels of spans in a parsing tree.
> Because there is no score for each token in our model architecture, there may be a gap to apply conventional interpretable metrics to our model. e.g. there is no straightforward metric to rank top K tokens.
> Nevertheless, we evaluated the interpretability of our model by performing an experiment based on the suggested paper (https://arxiv.org/abs/2111.07367) and reported it in Appendix A.4 in our revised paper. We did not include all the baselines due to time constraints, but the experiment shows the faithfulness of our span labels to a certain extent.
>
> Thanks for your help in improving our paper and we sincerely wait for your feedback.

---

> > ### Comment · Reviewer_Vws4 · 2022-11-30
> > **Response to revised paper**
> >
> > The revised paper is significantly clearer than the original. It's still not that clear, and I still think the paper's biggest weakness is its lack of clarity, but I think I can follow the derivation in 3.2.
> >
> > I am concerned about the conditional independence assumption made in equation (2). From what I understand, it's assuming something like "given a specific parse tree t, the quantity **number of possible label trees whose yield is a superset of the gold label set tau** is independent of the quantity **number of possible label trees whose yield does not contain any elements of $\mathcal{O}$**, where $\mathcal{O}$ is basically the set of all possible incorrect labels for the given sentence." It seems to be assuming that whether a given parse tree's yield provides ALL the correct labels is independent of whether its yield provides ANY incorrect labels. This seems pretty questionable. Though admittedly, machine learning is full of methods that shouldn't work in theory, but do in practice.
> >
> > Regardless of whether my understanding is correct, I think a few sentences conceptually explaining the independence assumption and when it does and doesn't hold would improve the paper.
> >
> > I also note that there are quite a few grammatical errors. E.g.,
> > - "Let $\mathcal{F}$ denote the set union of all the task labels and {$\phi_T, \phi_{NT}$}**, $\mathcal{O}$ denote**..." should be "Let $\mathcal{F}$ denote the set union of all the task labels and {$\phi_T, \phi_{NT}$}**, and let $\mathcal{O}$ denote**..."
> > - "node $n_{i,j}$ needs to be associated with either $\phi_{NT}$ or $\mathcal{l}$, whose **probability is**" should be "node $n_{i,j}$ needs to be associated with either $\phi_{NT}$ or $\mathcal{l}$, whose **probabilities are**"
> >
> > I'd suggest verbally reading the paper out loud to yourself and pasting the text portions of the paper's LaTeX file into a grammar checker such as Grammarly.
> >
> > I also appreciate the authors' inclusion of the interpretability experiment in A.4, and the results there seem fairly strong.
> >
> > All in all, I've revised my score up to a 6, primarily reflecting the improved clarity of the paper, my improved understanding of what the paper's method allows (span level classification without requiring span level labels), and the results in A.4. Concerns regarding the paper's clarity and the validity of the independence assumption remain and prevent me from assigning a higher score.

---

> > > ### Author Response · Authors · 2022-12-01
> > > **Thank you for your update**
> > >
> > > Many thanks for your approval of our supplemental experiment.
> > >
> > >
> > > Regarding the independence assumption, your understanding of the independence assumption is correct and we totally agree with you that we cannot prove or falsify it theoretically. It assumes "whether a given parse tree's yield provides ALL the correct labels is independent of whether its yield provides NONE incorrect labels. "
> > >
> > > To demonstrate that introducing the independence assumption has no negative impact, we include a baseline model based on the dynamic algorithm that is not based on the independence assumption. As shown in Table 1, S.N.$_{fp}$ denotes the model based on the DP algorithm with exponential complexity which is described in Appendix A.1. The results show the models with the independence assumption are very close to the models without the assumption. So empirical results show it works.
> > >
> > > On the other side, we argue the independence assumption used in our objective actually is weaker than the one used in conventional multi-label classification tasks. Formally, conventional multi-label classification is the problem of finding a model that maps inputs $\textbf{x}$ to binary vectors $\textbf{y}$; that is, it assigns a value of 0 or 1 for each element (label) in $\textbf{y}$. So the objective of multi-label classification is to minimize: $-log P(\bigcap_{i \in \mathcal{T}}^{} y_{i}=1,\bigcap_{j \in \mathcal{O}}^{} y_{j}=0|x)$, where $\mathcal{T}$ denotes the indices for golden labels and $\mathcal{O}$ denotes the indices not in $\mathcal{T}$. It's impossible to tractably estimate it without introducing some conditional independence assumption. By assuming conditional independence, we have:
> > > \begin{equation}
> > > P(\bigcap_{i \in \mathcal{T}}^{} y_{i}=1,\bigcap_{j \in \mathcal{O}}^{} y_{j}=0|x) \approx P(\bigcap_{i \in \mathcal{T}}^{} y_{i}=1 |x) \cdot P(\bigcap_{j \in \mathcal{O}}^{} y_{j}=0|x)
> > > \end{equation}
> > > \begin{equation}
> > > \log P(\bigcap_{i \in \mathcal{T}}^{} y_{i}=1 |x) \approx \log \prod_{i \in \mathcal{T}}^{}P(y_{i}=1|x)=\sum_{i \in \mathcal{T}}^{} \log P(y_i=1|x)
> > > \end{equation}
> > > \begin{equation}
> > > \log P(\bigcap_{j \in \mathcal{O}}^{} y_{j}=0 |x) \approx \log \prod_{j \in \mathcal{O}}^{}P(y_{j}=0|x)=\sum_{j \in \mathcal{T}}^{} \log P(y_j=0|x)
> > > \end{equation}
> > > which could finally be reformulated to the well-known binary cross entropy loss $-\sum_{i}^{}\hat{y}_i\log y_i + (1 - \hat{y}_i) \log (1 - y_i)$, where $\hat{y}$ is the ground truth and $y$ is the output probability of a model.
> > >
> > > The logic of equation(2) in our revised paper is similar to the above equations (1)(2)(3). $P(\hat{t}^{[\mathcal{T} \subseteq \mathcal{Y}(\hat{t})]}|t)$ is equivalent to $P(\bigcap_{i \in \mathcal{T}}^{} y_{i}=1 |x)$ and $P(\hat{t}^{[\mathcal{O} \cap \mathcal{Y}(\hat{t}) \ne \phi]}|t)$ is equivalent to $P(\bigcap_{j \in \mathcal{O}}^{} y_{j}=0|x)$.
> > > So it's something like  "whether a given parse tree's yield provides ALL the correct labels is independent of whether its yield provides NONE incorrect labels. "
> > > But we don't require the independence assumption to estimate the latter. As you mentioned, it couldn't be proven theoretically but just works in conventional multi-label classification. We argue our weaker version of the independence assumption also empirically works.
> > >
> > > Thanks again for your feedback!

---

### Author Response · Authors · 2022-12-02
**Author response**

We thank all the reviewers for their reviews and professional suggestions. We have improved our paper thanks to their feedback.

As we understand it, the reviewer's main concern is lack of clarity.
We believe the following limited revisions would further improve the clarity of our paper and address the reviewers' concerns in their new comments:

1) Add a "Model" section between the original Sections 3.1 and 3.2,  meanwhile merge the original Section 3.3(top-down encoder) to the new section. Clearly define the components and parameters in the Symbolic-Neural model as follows:

There are three components in the Symbolic-Neural model:

    A structured LM backbone Fast-R2D2 which is used to parse a sentence to a binary tree with node representations.
    An MLP which is used to estimate the label distribution of each node.
    A top-down encoder as an optional module which allows inner nodes to acquire contextual information.

We denote the parameters used in the Structured LM as $\Phi$ and the parameters used in the MLP layer and the top-down encoder as $\Theta$.

Thus in the training objective section,$L_{cls}^{t}$ and $L_{self}$ could be written as $L_{cls}^{t}(\Phi,\Theta)$ and $L_{self}(\Phi)$ to clearly specify what parameters are being optimized during training.

2) We will add a few sentences to explain the loss and the independence assumption.
For the loss, we will literally explain that it aims to maximize the probability of a given tree $t$ transferring to a label tree $\hat{t}$ yielding labels that are consistent with the ground-truth labels.
For the independence assumption, we will explain that we assume that the states of labels are independent of each other, where the state of a label indicates whether the label is contained in the yield result. Actually, such an assumption is consistent with the independence assumption in conventional multi-label classification as we explained in our 2nd response to Reviewer Vws4.


Meanwhile, we will release our source code, which will also help readers understand the whole work.

---

### Decision · Program_Chairs · 2023-01-20

**Decision:**

Accept: poster

**Justification For Why Not Higher Score:**

Writing quality of the paper can be improved.

**Justification For Why Not Lower Score:**

I think this paper presents an interesting approach that provides explicitly interpretable text classification.

**Metareview: Summary, Strengths And Weaknesses:**

This paper addresses the problem of interpretable text classification. It leverages pretrained structured language model to generate parse tree and introduces a training objective that allows for prediction at individual nodes of the parse tree (span-level prediction) while trained on only sentence-level labels. Because the prediction is done at span level, it offers natural explanation for the prediction in the form of text spans. The model is evaluated on several text classification tasks (at sentence level) and its prediction performance is comparable to fined tuned BERT and Fast-R2D2. The paper also evaluates the model's explanation against human provided rationale and show that the proposed method's explanation is well aligned with human rationale.
b) Strengths:
Self-explaining text classification is an interesting and important topic.
The training objective is novel, and the results on explanation are strong, highlighting the capability of the model to provide span level prediction/explanation with only sentence level supervision.
c) Weakness:
There is no clear benefit in terms of prediction performance.
The writing of the paper still has some issues but could be fixed. The first half of the paper has been substantially improved based on reviewers' comments. But the second half still needs work. The experiments section is dense and contains too much information and many cross references to the appendix, making it difficult to follow, and distill the key messages.

**Note From Pc:**

if the above contains the word "oral" or "spotlight" please see: "oral" presentation means -> notable-top-5% and "spotlight" means -> notable-top-25%. As stated in our emails, we are disassociating presentation type from AC recommendations

**Summary Of Ac-Reviewer Meeting:**

This was not initially identified as a borderline paper but later evolved into one. I had conversation with one of the reviewers to discuss the readability issue of the paper, which was substantially improved after the revision, but still has some (potentially fixable) issues.